# Classification of Faults Operation of a Robotic Manipulator Using Symbolic Classifier

Nikola Anđelić *, Sandi Baressi Šegota, Matko Glučina and Ivan Lorencin

Faculty of Engineering, University of Rijeka, Vukovarska 58, 51000 Rijeka, Croatia
* Correspondence: nandelic@riteh.hr

**Abstract:** In autonomous manufacturing lines, it is very important to detect the faulty operation of robot manipulators to prevent potential damage. In this paper, the application of a genetic programming algorithm (symbolic classifier) with a random selection of hyperparameter values and trained using a 5-fold cross-validation process is proposed to determine expressions for fault detection during robotic manipulator operation, using a dataset that was made publicly available by the original researchers. The original dataset was reduced to a binary dataset (fault vs. normal operation); however, due to the class imbalance random oversampling, and SMOTE methods were applied. The quality of best symbolic expressions (SEs) was based on the highest mean values of accuracy ($\overline{ACC}$), area under receiving operating characteristics curve ($\overline{AUC}$), $\overline{Precision}$, $\overline{Recall}$, and $\overline{F1-Score}$. The best results were obtained on the SMOTE dataset with $\overline{ACC}$, $\overline{AUC}$, $\overline{Precision}$, $\overline{Recall}$, and $\overline{F1-Score}$ equal to 0.99, 0.99, 0.992, 0.9893, and 0.99, respectively. Finally, the best set of mathematical equations obtained using the GPSC algorithm was evaluated on the initial dataset where the mean values of $\overline{ACC}$, $\overline{AUC}$, $\overline{Precision}$, $\overline{Recall}$, and $\overline{F1-Score}$ are equal to 0.9978, 0.998, 1.0, 0.997, and 0.998, respectively. The investigation showed that using the described procedure, symbolically expressed models of a high classification performance are obtained for the purpose of detecting faults in the operation of robotic manipulators.

**Keywords:** genetic programming; oversampling methods; robot fault operation; random oversampling; symbolic classifier; SMOTE





## 1. Introduction

The main mode of operation for industrial robotic manipulators is unsupervised automatic operation. In modern production facilities, the industrial robots will perform the set tasks completely autonomously, either repeating the same series of operations [1] or adjusting the movement according to sensor inputs [2]. One of the issues that can arise with such an operation is faults. For this paper, faults are in this instance catastrophic faults that occur when the industrial manipulator hits an object which existed in its surroundings [3]. Such a failure requires an immediate ceasement of operation—as further movement can cause significant damage to the equipment or, crucially, a more serious injury to the human operator. One of the ideas to achieve a timely stopping is to utilize machine learning (ML) methods.

ML methods have a wide application in the area of fault detection. In [4], the authors discuss the application of ML algorithms in the Internet of Things (IoT) systems to detect possible faults in photovoltaic systems. Authors conclude wide applicability of such techniques can lead to promising results. The authors in [5], have utilized Long short-term memory (LSTM) networks on the time series data relating to the faults in power transmission. The developed models are tested and demonstrate high resiliency despite varying operating conditions. In [6], the authors demonstrate the use of ML in a maritime application—namely, for the detection of faults and operation optimization. The one-dimensional convolutional neural networks (CNN) authors use show promising results.

Dang et al. [7] demonstrate the application of multiple algorithms on the problem of arcing fault detection in DC systems. The authors compare long short-term memory (LSTM), gated recurrent unit (GRU), and deep neural network (DNN) algorithms, concluding that all have satisfying performance. Another application in electrical engineering is demonstrated in [8]. The authors demonstrate an application of private reinforcement learning to achieve a precise detection that is above the stated baseline. In [9], the ANN and k-nearest neighbor approaches were applied to the problem of spiral bevel gear fault detection. In [10], the method was proposed for effective connection between users and providers. The method can discover users' suitable queries and understand their preferences. The proposed system anticipates the connections among various technical fields and helps personnel discover useful technical knowledge. A combination of MC-SIOT with deep learning algorithms can better maintain the functionality system state. In [11], chaotic back propagation (BP) neural network was utilized for the prediction of smart manufacturing information system reliability. The results showed that when SMIS fails, the failure behavior can easily lead SMIS into chaos through the propagation of an interdependent network.

In the previous literature overview, it is shown that different algorithms outperform others depending on operating conditions, indicating the need for testing novel algorithms on fault detection problems. From the previously presented literature, it is obvious that such methods may be easily applicable to robotic fault detection.

In [12], the experimental investigation on a robot manipulator is presented in which the neural network was used to analyze the vibration condition on joints. In this research, two different types of neural networks were used, i.e., self-organizing map neural network (SOMNN) and radial basis neural network (RBNN). The investigation showed that at both running speeds, the RBNN outperforms the SOMNN with a *RMSE* value of 0.0004. The fault diagnosis approach of robotic manipulators was investigated in [13] using support vector machines (SVM). Using the radial basis function (RBF) network the interpolation of unknown actuator faults was achieved. The interpolation of unknown actuator faults was achieved using a radial basis function (RBF) network and was successfully tested experimentally. A neural-network fault diagnosis module and a reinforcement learning-based fault-tolerant control module were used in [14] as fault-tolerant control frameworks for robotic manipulators that are subjected to joint actuator faults. After the actuator fault is detected and diagnosed, the additive reinforcement learning controller will produce compensation torques to achieve system safety and maintain control. Using the proposed method an average accuracy of 97% was achieved. The SVM was used in [15] for fault detection of the robot manipulator and the results were compared with results obtained with ANN. The results showed that the recognition rate was higher in the case of SVM (99.6%). An adaptive neural network model for diagnosing faults (FD) in combination with adaptive, fuzzy, backstepping, variable control (FC) was proposed and used in [16] for fault-tolerant control. The variable structure observer (VSO) was used for the FD technique of the robot manipulator while higher-order VSO (HVSO) was used to solve the chattering phenomenon of VSO. The estimation performance of HVSO was improved by the implementation of a neural network in the FD pipeline. The improvement of FC was achieved using adaptive higher-order variable structure observer (AHVSO) and the results showed 27% to 29% improvement in faults detection when compared to HVSO and VSO methods.

The faults in sensors and actuators of the scara robot were detected using ANN and fuzzy logic in [17]. The proposed approach successfully detects and isolates the actuator and sensor faults.

The scientific papers in which the same dataset [18] was used as in this research are listed and described below. The self-organizing map (SOM), and genetic algorithm with SOM (GA-SOM) have been used in [19] to detect the fault operation of the robot manipulator. The GA-SOM outperforms the SOM with an achieved classification accuracy of 91.95%. In [20], the performance of base-level and meta-level classifiers were compared on [18]. The ML classifiers that have been used were Naive Bayes, boosted Naive Bayes,

bagged Naive Bayes, SVM, boosted SVM, bagged SVM, decision table (DT), boosted DT, bagged DT, decision tree (DTr), boosted (DTr), bagged DTr, plurality voting, stacking meta decision trees, and stacking ordinary decision trees. The bagged Naive Bayes method achieved the highest classification accuracy of 95.24%. The deep convolutional neural networks (DCNN) are used in [21] to detect robot manipulator execution failures using sensor data from force and torque sensors. The best classification accuracy of 98.82% was achieved with one-dimensional CNN followed by two-dimensional CNN with 98.77% classification accuracy. In [22], 24 neural network (NN) architectures with seven learning algorithms were used to predict execution failures. The 10-8-5-4 NN architecture with the Bayesian regularization algorithm achieved a classification accuracy of 95.45%. The multi-layer perceptron (MLP) was used in [23] to predict robot execution failures. The highest classification accuracy with MLP was 90.45%. The deep-belief neural network (DBN) for detection of the robotic manipulator's failure execution was investigated in [24]. The performance of DBN was compared with other standard ML classifiers (C-support vector classifier, logistic regression, decision tree classifier, K-Nearest Neighbor Classifier, MLP, AdaBoost Classifier, random forrest classifier, bagging classifier, voting classifier) and results showed that the approach has a higher detection accuracy (80.486%) than other algorithms. In [3], the MLP, SVM, CNN, and Siamese neural network (SNN) were utilized for the detection of faults during the operation of the robotic manipulator. The highest $F1 - Score$ value (1.0) was achieved with SNN.

The representation of all research papers that used the dataset [18] in their investigation with various ML algorithms and achieved classification accuracy ($ACC$, $F1 - score$) values are presented in Table 1.

**Table 1.** The list of research papers, with described ML methods and achieved results in fault operation obtained using dataset [18].

| Reference | Methods | Results |
|:---:|:---:|:---:|
| [19] | GA-SOM, SOM | $ACC$ : 91.95% |
| [20] | Naive Bayes, Boosted Naive Bayes, Bagged Naive Bayes, SVM, Boosted SVM, Bagged SVM, Decision Table (DT), Boosted DT, Bagged DT, Decision Tree (DTr), Boosted (DTr), Bagged DTr, Plurality Voting, Stacking Meta Decision Trees, Stacking Ordinary Decision Trees | $ACC$ : 95.24% |
| [21] | DCNN | $ACC$ : 98.82% |
| [22] | NN with Bayesian regularization | $ACC$ : 95.45% |
| [23] | MLP | $ACC$ : 90.45% |
| [24] | DBN, C-support vector classifier, logistic regression, decision tree classifier, K-Nearest Neighbor Classifier, MLP, AdaBoost Classifier, Random Forrest Classifier, Bagging Classifier, Voting Classifier | $ACC$ : 80.486% |
| [3] | SNN | $F1 - Score$ : 1.00 |

It can be seen from Table 1 that in none of the research papers was the genetic programming-symbolic classifier (GPSC) used.

*The Description of Research Novelty, Investigation Hypotheses with Overall Scientific Contribution*

It can be noticed from the previously presented state of the art, that there are a few papers in which ML algorithms were used for fault operation analysis and detection. The main disadvantage which can be noticed in the previous literature overview is that these ML models are computationally intensive, i.e., they require large computational resources, especially the CNN or DNN. After these algorithms are trained on a specific dataset they have to be stored to be used in the future for processing new data. Storing and re-using these models requires a lot of computational resources so their usage in systems that control the robot manipulators is questionable. Another problem with the previously mentioned algorithms is that it is nearly impossible to express the complex models obtained by them as comparatively simple mathematical expressions which are understandable to scientists and engineers.

The research novelty is to present the process of the GPSC algorithm implementation to obtain symbolic expressions (SEs) which can detect fault operation with high classification performance. The SEs are easier to implement in existing control systems of robotic manipulators since they have lower computation-wise performance requirements when compared to complex models such as CNN or DNN. The main idea of this research is the procedure presentation of the GPSC algorithm application to generate SEs which will be utilized as high-performance detection models of robot manipulator faults.

The algorithm used in this paper [25] begins its process by creating the population members of the initial population that are not able to solve a particular task. However, through the predefined number of generations with the application of evolutionary computing operations (recombining and mutating), they adapt as the solution for a particular task.

From state-of-the-art, defined novelty, and research ideas, the following hypotheses will be investigated in this paper:

- Is there a possibility to use the GPSC algorithm to generate SEs for the detection of fault operation of a robotic manipulator with high classification performance?
- Can the proposed algorithm achieve high classification performance on datasets that are balanced using various oversampling methods?
- Is it possible to use the GPSC algorithm with a random selection of hyperparameter values (RSHV) method, validated with 5-fold cross-validation (5FCV) to obtain SEs for the detection of fault operation of a robotic manipulator with high classification accuracy?
- Can the high performance of SEs that consist of a reduced number of input parameters be achieved?

The scientific contributions are:

- Investigates the possibility of obtaining a SE for robot manipulator fault operation using GPSC algorithm.
- Investigates the influence of dataset oversampling methods (OMs) on (SEs) classification performance.
- Investigates if using the GPSC algorithm with an RSHV method, validated using a 5FCV process can generate a set of robust SEs with a high detection accuracy of robot manipulator fault operation.

The outline of this paper consists of four sections. Firstly, the Materials and Methods section where research methodology, dataset, OMs, GPSC, RSHV, 5FCV procedure, and resources are described. Afterward, in the Results section, the SEs obtained using the GPSC algorithm with RSHV validated with 5FCV are presented. In the Discussion section, the results are discussed and justified in detail. Lastly, in the Conclusions section, the conclusions of the conducted investigation are provided.

## 2. Materials and Methods

The approach used in this investigation is described here, starting with the Dataset description and accompanying statistical analysis, oversampling methods, GSPC algorithm, model evaluation, and finally the computational resources used during the research.

### 2.1. Research Methodology

From the initial investigation, it was noticed that the dataset is highly imbalanced. For that reason, it was necessary to apply balancing methods to equalize the number of samples per class. This was achieved using random oversampling and SMOTE methods. The aforementioned methods generated two variations of the initial dataset and were used in the GPSC to obtain SEs. The results obtained on each modified dataset were compared, and the one with the highest classification performance and the smallest size is considered to be the best—as the goal is to obtain the best performing, but still relatively simple, equations. The final validation is performed on the initial dataset. The presentation of this process is given in Figure 1.

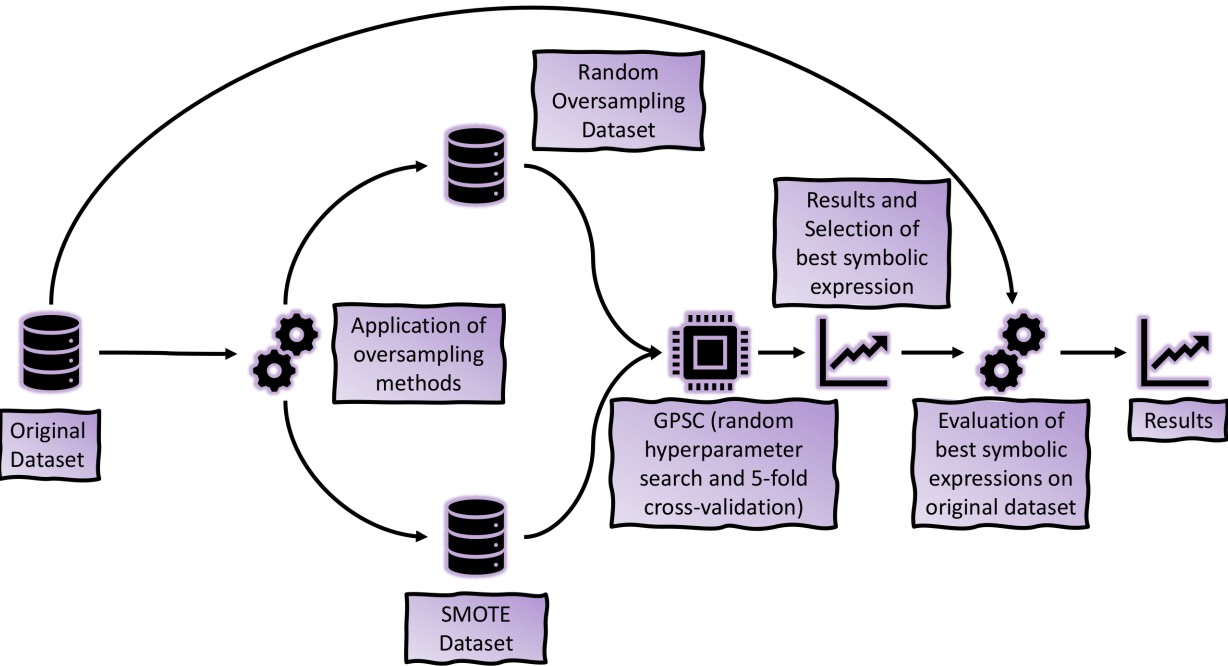

**Figure 1.** The methodology research flowchart.

### 2.2. Dataset Description

The publicly available dataset [18] was used in this research. The dataset is well documented in [26,27] so in this subsection, only a brief dataset description is given. The dataset consists of a total of 463 data points and each of the data points $D$ is shaped as:

$$D = \begin{bmatrix} F_1^x & F_1^y & F_1^z & T_1^x & T_1^y & T_1^z \\ F_2^x & F_2^y & F_2^z & T_2^x & T_2^y & T_2^z \\ \vdots & \vdots & \vdots & \vdots & \vdots & \vdots \\ F_{15}^x & F_{15}^y & F_{15}^z & T_{15}^x & T_{15}^y & T_{15}^z \end{bmatrix}. \tag{1}$$

As seen in the equation above, each of the data points consists of 90 measurements, organized into 15 subpoints. Each of the subpoints is a measurement of force and torque in each of the $x$, $y$, and $z$ axes. The time difference between each point is 0.315 s, meaning that

the entirety of the data point consists of 28.35 s. The statistical analysis and GPSC variable representation for all dataset variables are shown in Table A1 (Appendix A.1).

It should be noted that in GPSC all input variables are represented with $X_i$ notation where $i = 0, ..., 89$ since there are 90 input variables. The output variable in GPSC is represented with $y$.

The data points are split into 15 classes in total—"collision in tool", "collision in part", "bottom obstruction", "bottom collision", "lost", "moved", "slightly moved", "right collision", "left collision", "back collision", "obstruction", "front collision", "collision", "normal", and "ok". The number of elements in each class is given in Figure 2.

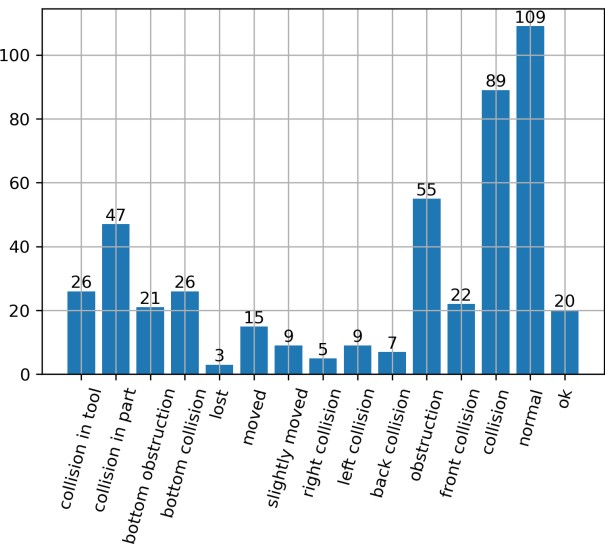

**Figure 2.** The distribution of the data points in the dataset, by class.

Out of the listed classes, "normal" and "ok" indicate a non-faulty/normal operation, while the others indicate fault. Since for a lot of applications just knowing whether the fault has occurred is enough, and it is not necessary to immediately discern the type of it, the dataset can be made into a binary dataset by combining all the "faulty" data points and all the "non-faulty" data points, as shown in the Figure 3.

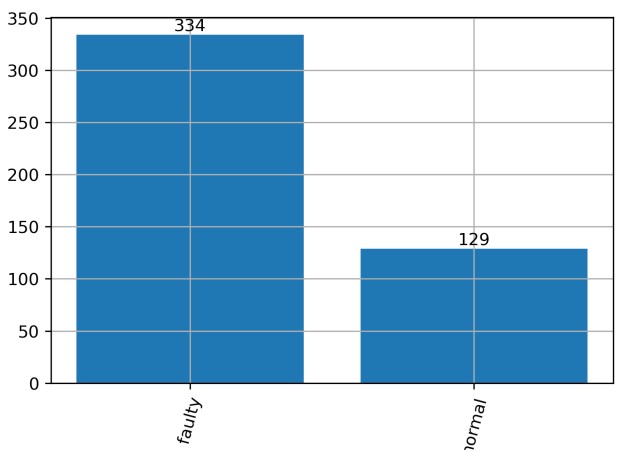

**Figure 3.** The distribution of the data points in the dataset, by class, after sorting for binary classification.

As can be seen from Figures 2 and 3, the classes in the dataset are not balanced. This fact opens the possibility of applying the dataset balancing techniques which will be described in the following section.

In this paper, only the binary problem will be considered because even balancing problems could not provide sufficient datasets in the case of multiple classes. In a multi-class problem, as seen in Figure 2, there are 15 classes. The extremes in the number of samples are lost class with only 3 samples and normal class with 109 samples. An equal number of samples per class (14 classes in total) can be achieved using OMs. However, even with the implementation of dataset balancing methods, each class would contain only 109 samples which is not enough for the implementation of ML methods.

One of the investigations important to statistical dataset analysis is the correlation analysis between dataset variables. The correlation between variables could indicate if the dataset used in the training process will produce the trained ML model with high classification accuracy or not. In this paper, authors calculated Pearson's coefficients of correlation which return the values in the range of $< -1.0, 1.0 >$. The strong correlation ranges are in the range of $\pm < 0.5, 1.0 >$. The low correlation range is considered to be $< -0.5, 0.5 >$, where 0 is the weakest correlation. If the value is negative, the growth of the variables is inverse, and vice-versa. Since the dataset consists of 91 variables and the correlation heatmap is too large for better visualization the interrelationship between dataset variables is presented in Figure 4.

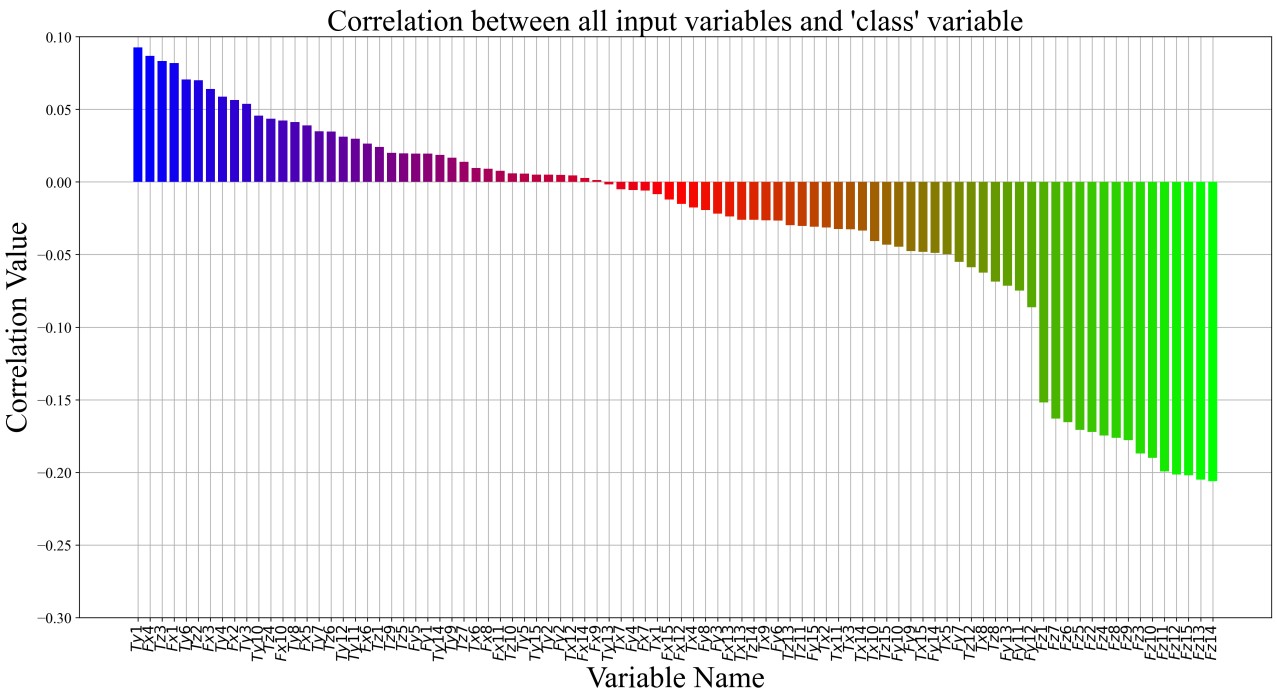

**Figure 4.** Correlation of the output ("Class") variable and the inputs.

As seen in Figure 4, correlation values between the input and the target variables are in the 0.1 to $-0.21$ range. However, the majority of variables (75) are weakly correlated with the target variable ("Class" with a 0.05 to $-0.1$ range). Due to low correlation values between dataset variables, all variables were used in the GPSC algorithm to obtain SEs. In the Results section, the analysis of obtained SEs is performed, and analyzed which input variables ended up in the best SEs.

### 2.3. Dataset Balancing Methods

Due to the low sample number per class even in the binary classification problem (Figure 3) only dataset OMs were used and these are: random oversampling and SMOTE methods. The class that contains a low number of points in the dataset is labeled as a "minority" class. On the other hand, the opposite is referred to as the "majority" class.

2.3.1. Random Oversampling

Oversampling data randomly is one of the simplest balancing methods in which the minority class samples are randomly selected and copied to match the number of majority class samples.

2.3.2. SMOTE

The Synthetic Minority Oversampling Technique (SMOTE) [28] is a technique in which synthetic samples are generated in the following way:

- Calculate the amount of samples $N$ that have to be generated to obtain 1:1 class distribution;
- Application of iterative process consisting of the following steps:
  - A random selection of minority class sample is performed;
  - K nearest neighbors (by default K = 5), are searched for;
  - The $N$ of $K$ samples are randomly chosen to generate new instances using the interpolation procedure. The difference between the sample under consideration and selected neighbors is used and increased by a factor in the range of $< 0, 1 >$, which is appended to the sample. Using this procedure new synthetic samples are created.

Using the previously described procedure new synthetic samples are created along the line segments which can be used to connect two dataset samples. The results of the balancing dataset using different oversampling methods are listed in Table 2.

**Table 2.** The results of dataset balancing methods.

| Dataset Balancing Method Name | Number of Minority Class Samples | Number of Majority Class Samples | Total Number of Samples |
|---|---|---|---|
| Random Oversampling | 334 | 334 | 668 |
| SMOTE | 334 | 334 | 668 |

*2.4. Genetic Programming Symbolic Classifier*

The GSPC algorithm is a method in which a randomly generated initial population is unable to detect the target variable and through the evolution process makes them fit for detection of the target variable with high classification accuracy. When GPSC execution begins the initial population is created. This is a complex process that requires the definition of specific GPSC hyperparameter values, i.e., NumGens, PopSize, InitDepth, InitMethod, TourSize, ConstRange, and FunSet. To define input variables before GPSC execution the train part of the dataset must be provided. The InitMethod used to create the initial population in this paper was "ramped half-and-half". This method combines the full and grow method and to ensure population diversity the maximum depth limit (InitDepth) is defined in a specific range. The length of the SE (population member) is measured with the number of elements, i.e., constants, functions, and input variables. It should be noted that in the results section, the best SEs will also be compared in terms of length.

To create one population member, input variables from the dataset are required as well as functions and constants. The constants are numbers defined in range using the hyperparameter ConstRange. The functions are randomly selected from the FunSet and in this case, the FunSet consists of the following functions: "$+$", "$-$", "$\cdot$", "$\div$", "$min$", "$max$", "$sin$", "$cos$", "$tan$", "$ln$", "$log_2$", "$log_{10}$", "$\sqrt{}$". After the creation of the initial population, the next step is to evaluate and calculate the fitness value of population members. In this paper the logarithmic loss fitness function was used, which is calculated as:

- With the obtained expression, model the predicted output class for all training data points;

- Calculate the Sigmoid function of the generated output:

$$S(t) = \frac{1}{1 + e^{-t}},$$ (2)

- The log-loss function is calculated with the predicted and real training data points, per:

$$H_p(q) = -\frac{1}{N} \sum y_i \cdot \log(p(S_i)) + (1 - y_i) \cdot \log(1 - p(S_i)),$$ (3)

with $y$ representing the real dataset output and $p(S)$ is the output of the sigmoid function.

In GPSC two different genetic operators (GOs) are used, crossover and mutation, with mutation split into three different subtypes: subtree mutation, hoist mutation, and point mutation. These four genetic operations are performed on winners of tournament selection. In the case of crossover, two winners of tournament selection are required. The hyperparameter names, using which the probabilities of GOs are defined, are Cross, SubMute, HoistMute, and PointMute. The sum of all genetic operation probabilities can be equal to less than 1. If it is less than 1 then some parents enter the next generation unmodified.

The GPSC has two termination criteria, i.e., StopCrit (lowest predefined value of the fitness function) and NumGens (maximum number of generations). The MaxSamp hyperparameter defines how much of the training dataset will be used to evaluate the population members from generation to generation.

To prevent the bloat phenomenon the parsimony pressure method is used. The bloat occurs during GPSC algorithm execution when the population members' size rapidly increases from generations without benefiting fitness function. When this method is used, the value of ParsCoef must be defined. This method is introduced during the tournament selection process when very large population members are found. To make them less favorable for winning tournament selection their fitness value is modified. This is achieved by using the equation:

$$f_p(x) = f(x) - cl(x),$$ (4)

where $l(x)$ is the size of the unit's expression, $c$ is the parsimony pressure coefficient and $l(x)$ is the total length of the population member.

### 2.5. Training Procedure of the GPSC Algorithm

To develop an RSHVs method the initial tuning of the GPSC hyperparameters had to be performed. The hyperparameter ranges are listed in Table 3.

The ranges in Table 3 were defined through the initial tuning of the GPSC algorithm. The PopSize was set to the 100–1000 range that was propagated for 100–300 NumGens. The StopCrit range values are very low to prevent early termination of GPSC algorithm execution. The GO coefficients have the same ranges from which the values are randomly selected. The idea is to see which one of the genetic operations will be most influential in case of obtaining the set of best SEs. The MaxSamp was set to the 0.99–1 range, i.e., the entire training dataset was used to evaluate each population member in each generation. The initial investigation showed that the best range for this research is the one shown in Table 3 since it will prevent the bloat phenomenon while enabling stable growth of population members while lowering the fitness function value.

The modeling using the described algorithm consists of:

- Random hyperparameter selection;
- Training the GPSC algorithm with the randomly selected hyperparameters;
- Evaluating obtained SEs and testing if all EMs are above 0.99. If they are above 0.99 the process is terminated, otherwise, the process is repeated.

**Table 3.** The range of GPSC hyperparameters.

| GPSC Hyperparameter | Range |
|---|---|
| PopSize | 100–1000 |
| NumGens | 100–300 |
| TourSize | 100–300 |
| InitDepth | 3–12 |
| Cross | 0.001–1 |
| SubMute | 0.001–1 |
| HoistMute | 0.001–1 |
| PointMute | 0.001-1 |
| StopCrit | $1 \times 10^{-7}$–$1 \times 10^{-6}$ |
| MaxSamp | 0.99–1 |
| ConstRange | $-10{,}000$–$10{,}000$ |
| ParsCoef | $1 \times 10^{-5}$–$2 \times 10^{-4}$ |

Some successful implementations of this process are well documented in [29–31]. Initially, the dataset is split in a 70:30 training/testing data split, because of faster execution. Then, the obtained models are tested using a 5FCV procedure on 70% of the dataset. The training procedure is repeated until it yields a score higher than 0.99. If that fails, the random hyperparameter choice, along the entire process, is repeated.

### 2.6. GPSC Evaluation Methods

In this subsection, the evaluation metrics (EMs) are described, as well as the process of evaluation methodology (EMT).

#### 2.6.1. Evaluation Metrics

The accuracy, area under received operating characteristics curve (AUC), recall and precision values, and F1-score were used to evaluate the obtained models. When discussing classification problems, there are four types of outcomes: true positive (TP), true negative (TN), false positive (FP), and false negatives (FN). The TP is an outcome where the ML model correctly predicts the positive class. In the case of binary classification, there are two classes defined as positive (class labeled as 1) and negative class (class labeled as 0 or $-1$). The TN is an outcome where the ML model correctly predicts the negative class. The FP is an outcome where the ML model incorrectly predicts the positive class. The FN is the outcome where the ML model incorrectly predicts the negative class.

The classification accuracy [32] is the fraction of prediction the ML model got right. The classification accuracy is defined as correct predictions and total values ratio. *ACC* can be written in the following form:

$$ACC = \frac{TN + TP}{FP + FN + TP + TN}. \tag{5}$$

*AUC* is one of the EMs used in this research that computes the area under the curve which defines the ratio of true positive and false negative predictions [33]. The precision metric value [34] gives information on how many positive classifications were correct, The *Precision* may be expressed as:

$$Precision = \frac{TP}{TP + FP}. \tag{6}$$

The recall metric value [34] provides information on how many actual members of the positive class were identified correctly by the trained ML model. The *Recall* is calculated as:

$$Recall = \frac{TP}{TP + TN} \tag{7}$$

The $F1 - Score$, according to [35], can be written as:

$$F1 - Score = \frac{2 \cdot Recall \cdot Precision}{Recall + Precision}. \tag{8}$$

All evaluation metric values are expressed in the range of $[0, 1]$ with the higher score representing better performance.

### 2.6.2. Evaluation Methodology

The process of determining the models starts with the random determination of hyperparameters. Then, the cross-validation is performed and after each fold, the evaluation metric values are calculated. After the cross-validation process is performed, the EMs are determined. If the values are higher than 0.99 then conclusive training and evaluation are performed. If the values of EMs are below the threshold the process is repeated.

In case the system progress to the final stage the GPSC algorithm training is executed on the training dataset part. After training is finished the results are evaluated on the training and testing dataset. If this yields satisfactory results, the process is finished. Otherwise, the process of repetition starts.

### 2.7. Computational Resources

The training and evaluation of the models are performed on the system as described below:

- Hardware
    - Intel i7-4770
    - 16 GB DDR3 RAM
- Software
    - Python 3.9.13
        * imblearn 0.9.1
        * scikit.learn 1.2.0
        * gplearn 0.4.2

## 3. Results

In this section, the results achieved on each balanced dataset and the best SEs evaluated on the initial dataset will be presented. In the subsection "The results obtained on balanced datasets using the GPSC algorithm" the best results are presented and compared. In the subsection entitled "Evaluating best models" the results of the best SEs evaluated on the initial dataset are presented.

### 3.1. The Results Obtained on Balanced Datasets Using the GPSC Algorithm

The hyperparameter values used to obtain the best SEs using the GPSC algorithm on each balanced dataset are shown in Table 4.

From Table 4, it can be seen that the best-performing model on the random oversampling dataset was obtained with a very small initial population (172) compared to the SMOTE case where the initial population consisted of 850 members. The population members in the case of random oversampling evolved for 293 (upper value of hyperparameter range) generations while in the SMOTE case the population members evolved for 227 (lower value hyperparameter range) generations. The SMOTE case had a higher tournament selection size, and the population members in tree form were much larger (6,11). In the case of random oversampling, the subtree mutation (0.49) was dominating genetic operation while in the SMOTE case the crossover (0.3) was the dominating one.

In both cases, as planned the stopping criteria were never met since the hyperparameter was very low. The parsimony coefficient in the SMOTE case was larger than in the random oversampling case. Classification performances are illustrated in Figure 5.

**Table 4.** GSPC hyperparameters yielding the best performing models on each dataset.

| Dataset Variation | GPSC Hyperparameters |
|---|---|
| Random Oversampling | 172, 293, 161, (4, 7), 0.1, 0.49, 0.36, 0.032, $7.78 \times 10^{-7}$, 0.99, $(-280.17, 5256.88)$, $6.95 \times 10^{-6}$ |
| SMOTE | 850, 227, 270, (6, 11), 0.39, 0.13, 0.23, 0.23, $1.4 \times 10^{-7}$, 0.99, $(-7689.72, 8984.85)$, $1.63 \times 10^{-5}$ |

The mean results and standard deviations, (error bars in Figure 5) are also given in Table 5.

**Table 5.** The evaluation metric values as well as computational time and the length of SE.

| Dataset Type | $\overline{ACC}$ $\pm SD(ACC)$ | $\overline{AUC}$ $\pm SD(AUC)$ | $\overline{Precision}$ $\pm SD(Precision)$ | $\overline{Recall}$ $\pm SD(Recall)$ | $\overline{F_1Score}$ $\pm SD(F_1Score)$ | Average CPU Time per Simulation [min] | Length of SEs |
|---|---|---|---|---|---|---|---|
| Random Oversampling | 0.985 $\pm 0.0056$ | 0.984 $\pm 0.0057$ | 0.9882 $\pm 0.00273$ | 0.982 $\pm 0.0086$ | 0.985 $\pm 0.0058$ | 100 | 729/368/78/119/159 |
| SMOTE | 0.99 $\pm 0.0089$ | 0.99 $\pm 0.00888$ | 0.992 $\pm 0.006$ | 0.9893 $\pm 0.0106$ | 0.99 $\pm 0.0084$ | | 544/430/354/387/334 |

Table 5 and Figure 5 show that SEs obtained in the SMOTE case have higher classification accuracy. The average CPU time in both cases is equal to 100 min. One of the factors that influence the CPU time to execute GPSC is the dataset size. The dataset is small in terms of samples (668 samples); however, the number of input variables for each sample is pretty large (90). The average CPU time to execute one iteration of 5FCV was 20 min so to perform the entire 5FCV is 100 min.

The average length of SEs obtained on a random oversampled dataset is lower than in the SMOTE dataset balanced case however, with the latter higher classification performance was obtained. So, the best results were those obtained on the SMOTE dataset.

## 3.2. Evaluating Best Models

As stated in the previous subsection the best model in terms of classification performance were those obtained on the SMOTE dataset. There are five equations, due to them being obtained in the 5-fold cross-validation. The best SEs are shown in Appendix B.

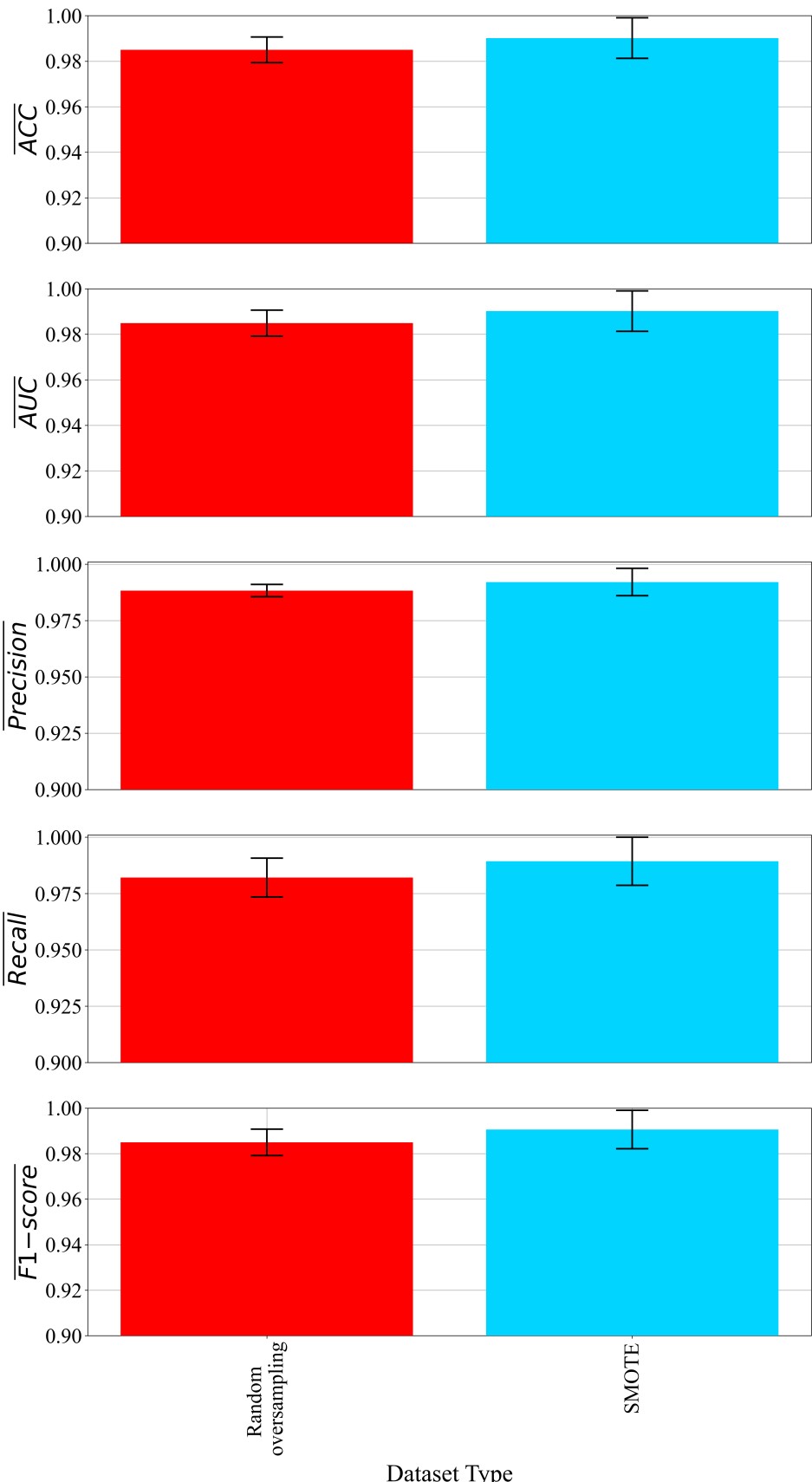

**Figure 5.** The mean and standard deviation values of EMs.

The Equation (A1) in Appendix B, consists of 19 input variables and these variables are $X_1$, $X_4$, $X_5$, $X_6$, $X_7$, $X_8$, $X_{11}$, $X_{12}$, $X_{16}$, $X_{18}$, $X_{40}$, $X_{44}$, $X_{48}$, $X_{56}$, $X_{60}$, $X_{77}$, $X_{78}$, $X_{81}$, and $X_{83}$. From Table A1 these input variables are $F_1^y$, $T_1^y$, $T_1^z$, $F_2^x$, $F_2^y$, $F_2^z$, $T_2^z$, $F_3^x$, $T_3^y$, $F_4^x$, $T_7^y$, $F_8^z$, $F_9^x$, $F_{10}^z$, $F_{11}^x$, $T_{13}^z$, $F_{14}^x$, $T_{14}^x$, and $T_{14}^z$. The Equation (A2) consists of 31 input variables and these variables are: $X_0$, $X_1$, $X_2$, $X_3$, $X_4$, $X_5$, $X_6$, $X_7$, $X_8$, $X_{14}$, $X_{15}$, $X_{16}$, $X_{18}$, $X_{19}$, $X_{24}$, $X_{25}$, $X_{28}$, $X_{30}$, $X_{39}$, $X_{44}$, $X_{45}$, $X_{47}$, $X_{49}$, $X_{52}$, $X_{58}$, $X_{63}$, $X_{73}$, $X_{77}$, $X_{78}$, $X_{80}$, $X_{88}$. From Table A1 these input variables are $F_1^x$, $F_1^y$, $F_1^z$, $T_1^x$, $T_1^y$, $T_1^z$, $F_2^x$, $F_2^y$, $F_2^z$, $F_3^z$, $T_3^x$, $T_3^y$, $F_4^x$, $F_4^y$, $F_5^x$, $F_5^y$, $T_5^y$, $F_6^x$, $T_7^y$, $F_8^z$, $T_8^x$, $T_8^z$, $F_9^y$, $T_9^y$, $T_{10}^y$, $T_{11}^x$, $F_{13}^y$, $T_{13}^z$, $F_{14}^x$, $F_{14}^z$, and $T_{15}^y$. The Equation (A3) consist of 23 input variables and these variables are $X_0$, $X_1$, $X_2$, $X_3$, $X_4$, $X_5$, $X_6$, $X_7$, $X_8$, $X_{18}$, $X_{28}$, $X_{36}$, $X_{40}$, $X_{43}$, $X_{49}$, $X_{59}$, $X_{68}$, $X_{69}$, $X_{70}$, $X_{81}$, $X_{82}$, $X_{84}$, $X_{86}$. From Table A1 these input variables are $F_1^x$, $F_1^y$, $F_1^z$, $T_1^x$, $T_1^y$, $T_1^z$, $F_2^x$, $F_2^y$, $F_2^z$, $F_4^x$, $T_5^y$, $F_7^x$, $T_7^y$, $F_9^y$, $F_9^y$, $T_{10}^z$, $F_{12}^z$, $T_{12}^x$, $T_{12}^y$, $T_{14}^x$, $T_{14}^y$, $F_{15}^x$, and $F_{15}^z$. The Equation (A4) consists of 28 input variables and these variables are $X_0$, $X_1$, $X_2$, $X_4$, $X_5$, $X_7$, $X_8$, $X_9$, $X_{16}$, $X_{18}$, $X_{20}$, $X_{21}$, $X_{22}$, $X_{25}$, $X_{27}$, $X_{28}$, $X_{40}$, $X_{44}$, $X_{46}$, $X_{49}$, $X_{50}$, $X_{51}$, $X_{59}$, $X_{76}$, $X_{77}$, $X_{79}$, $X_{80}$, and $X_{88}$. From Table A1 these input variables are $F_1^x$, $F_1^y$, $F_1^z$, $T_1^y$, $T_1^z$, $F_2^y$, $F_2^z$, $T_2^x$, $T_3^y$, $F_4^x$, $F_4^z$, $T_4^x$, $T_4^y$, $F_5^y$, $T_5^x$, $T_5^y$, $T_7^y$, $F_8^z$, $T_8^y$, $F_9^y$, $F_9^z$, $T_9^x$, $T_{10}^z$, $T_{13}^y$, $T_{13}^z$, $F_{14}^y$, $F_{14}^z$, and $T_{15}^y$. The Equation (A5) consists of 28 input variables and these variables are: $X_0$, $X_1$, $X_2$, $X_3$, $X_4$, $X_5$, $X_6$, $X_7$, $X_8$, $X_{14}$, $X_{16}$, $X_{18}$, $X_{24}$, $X_{27}$, $X_{28}$, $X_{32}$, $X_{33}$, $X_{36}$, $X_{42}$, $X_{46}$, $X_{56}$, $X_{63}$, $X_{66}$, $X_{67}$, $X_{69}$, $X_{73}$, $X_{74}$, $X_{86}$. From Table A1 these input variables are $F_1^x$, $F_1^y$, $F_1^z$, $T_1^x$, $T_1^y$, $T_1^z$, $F_2^x$, $F_2^y$, $F_2^z$, $F_3^z$, $T_3^y$, $F_4^x$, $F_5^x$, $T_5^x$, $T_5^y$, $F_6^z$, $T_6^x$, $F_7^x$, $F_8^x$, $T_8^y$, $F_{10}^z$, $T_{11}^x$, $F_{12}^x$, $F_{12}^y$, $T_{12}^x$, $F_{13}^y$, $F_{13}^z$, and $F_{15}^z$.

To compute the output Equation (A1) requires the lowest number of input variables (19) its length is lowest when compared to other equations. The Equations (A4) and (A5) require an equal number of input variables (28), however, not the same ones. Equation (A3) and Equation (A2) require 23 and 31 input variables, respectively.

Although it seems that these five SEs require a lot of input variables many of those variables are required in multiple SEs. Based on a detailed comparison of the best SEs it was found that a large number of variables are not required to compute the output. The input variables not in the best set of SEs are: $X_{10}$, $X_{13}$, $X_{17}$, $X_{23}$, $X_{26}$, $X_{29}$, $X_{31}$, $X_{34}$, $X_{35}$, $X_{37}$, $X_{38}$, $X_{41}$, $X_{53}$, $X_{54}$, $X_{55}$, $X_{57}$, $X_{61}$, $X_{62}$, $X_{64}$, $X_{65}$, $X_{71}$, $X_{72}$, $X_{75}$, $X_{85}$, $X_{87}$, and $X_{89}$. From Table A1, these input variables are $T_2^y$, $F_3^y$, $T_3^z$, $T_4^y$, $F_5^z$, $T_5^z$, $F_6^y$, $T_6^y$, $F_7^y$, $F_7^z$, $F_7^z$, $T_7^z$, $T_9^z$, $F_{10}^x$, $F_{10}^y$, $T_{10}^x$, $F_{11}^y$, $F_{11}^z$, $T_{11}^y$, $T_{11}^z$, $T_{12}^z$, $F_{13}^x$, $T_{13}^x$, and $F_{15}^y$, respectively.

The evaluation was performed as follows:

- Use the variables from the non-augmented dataset inside the expressions to obtain the predicted outputs;
- Apply the sigmoid function Equation (2) on that output, to transform the output of this function to an integer value;
- Compare the obtained values with the original target values from the dataset and obtain evaluation metric values.

The results of the evaluation of the models are listed in Table 6.

**Table 6.** The mean and standard deviation scores obtained using the best models on the non-augmented dataset.

| Evaluation Metric | Mean Value | Standard Deviation |
|---|---|---|
| *ACC* | 0.9978 | $5 \times 10^{-5}$ |
| *AUC* | 0.998 | $3.48 \times 10^{-5}$ |
| *Precision* | 1.0 | 0 |
| *Recall* | 0.997 | $1.6 \times 10^{-5}$ |
| *F1 − Score* | 0.9985 | $2.74 \times 10^{-5}$ |

Table 6 shows that the best model achieved even better classification performance on the original dataset. The dataset balanced with the SMOTE method did have some synthetic samples that deviate from the original dataset samples. The final evaluation of

the proposed approach would be to compare the results obtained on the original dataset with those results obtained in other research papers. In Table 7, the list of research papers with achieved classification performance is compared with results obtained in this research on the initial (binary) dataset.

As seen from Table 7, the highest score was achieved in the case of [3] using SNN. The results in this paper are slightly lower although this approach outperforms all other research papers in terms of classification accuracy. The additional benefit of using this approach is that a set of robust SEs was obtained which can be easily stored and used.

**Table 7.** The results comparison obtained in this paper with results from other research papers.

| Reference | Methods | Results |
|:---:|:---:|:---|
| [19] | GA-SOM, SOM | *ACC* : 91.95% |
| [20] | Naive Bayes, Boosted Naive Bayes, Bagged Naive Bayes, SVM, Boosted SVM, Bagged SVM, Decision Table (DT), Boosted DT, Bagged DT, Decision Tree (DTr), Boosted (DTr), Bagged DTr, Plurality Voting, Stacking Meta Decision Trees, Stacking Ordinary Decision Trees | *ACC* : 95.24% |
| [21] | DCNN | *ACC* : 98.82% |
| [22] | NN with Bayesian regularization | *ACC* : 95.45% |
| [23] | MLP | *ACC* : 90.45% |
| [24] | DBN, C-support vector classifier, logistic regression, decision tree classifier, K-Nearest Neighbor Classifier, MLP, AdaBoost Classifier, Random Forrest Classifier, Bagging Classifier, Voting Classifier | *AAC* : 80.486% |
| [3] | SNN | $F1 - Score$ : 100% |
| This paper | GPSC | *ACC* 99.78%, *AUC* 0.998%, *Precision* 100%, *Recall* 99.7% $F1 - Score$ 99.85% |

## 4. Discussion

The dataset consists of 90 input variables (forces and torques) and the output (target) variable which defines the operation class and there is a total of 15 classes. However, due to a small dataset and a large number of classes, the samples were divided into two classes "normal" and "fault" operation. So, by reducing the multi-class problem into a binary one the number of samples for these two classes increased. Since the initial dataset was improperly balanced, the balancing methods were applied to rectify this. One of the reasons why the multi-class problem was initially abandoned in this research is that some classes in the multi-class dataset had an extremely low number of samples per class (3, 5, 7, and 9). The dataset oversampling methods could not be applied to the classes with the extremely low number of original class samples. The initial investigation with GPSC using a multi-class dataset generated poor results so the multi-class problem was initially abandoned. Low correlation values were determined with the initial analysis, but all the variables ended up being included in the best-performing equations.

The definition of hyperparameter range for the random hyperparameter method is a time-consuming process since it requires initial tuning of hyperparameter ranges and

observing the GPSC algorithm execution behavior. The most crucial hyperparameters are genetic operation coefficients, the relation between population size and tournament selection size, and parsimony coefficient value. The starting hyperparameter tuning showed that an increase in the value of any genetic operation would not have any benefit towards the evolution process so they were all set to a pretty general case (all in the initial range of 0.001 to 1). Initial investigation showed that small tournament size and large population size can prolong GPSC execution time drastically. To prevent this the tournament size is, in the best-case scenario, 30% of the entire population. Finally, the most sensitive hyperparameter is the anti-bloat mechanism (parsimony). As mentioned, due to the low correlation between inputs and outputs, having a low parsimony coefficient value can result in a violent growth of equations. On the other hand, too large of a value will stifle the evolution process, resulting in the non-convergence of the model.

During this initial testing stage, it was ensured that the dominating stopping criteria would be a predefined maximum number of generations. The lowest value of the log loss fitness function was never achieved.

The described methodology was successful in generating models for all of the dataset variations, with the best-performing ones resulting from the SMOTE augmented dataset. The results shown in Table 5 show that random oversampling has lower classification accuracy than the SMOTE case and lower size of SEs. However, the size of SEs obtained in the SMOTE case is not so big so they were chosen as the best results.

The models trained on the SMOTE dataset showed that not all 90 inputs are necessary for classification. Each of these five SEs can be used to determine the class. However, all five are used to obtain the robust solution which was the initial idea of utilizing 5FCV. The SE that requires the lowest number of input variables is Equation (A1). This research also showed that not all of the input variables are required to obtain the best results. A detailed comparison of the best SEs showed that 26 input variables out of 90 are not needed to detect fault operation so these variables can be omitted from further investigation.

The utilization of the obtained models on the non-augmented dataset showed that using these SEs high classification accuracy could be achieved which can be seen from the results shown in Table 6.

## 5. Conclusions

The paper demonstrates an application of GSPC on a robot operation failures dataset. In the analyzed binary problem the GPSC was applied and validated for randomly selected hyperparameters, to obtain SE which can detect faulty operation. Since the binary variation of the original dataset was imbalanced the idea was to apply various balancing methods to equalize the dataset. The random oversampling and SMOTE methods were applied. The best results were evaluated on the initial dataset. After conducted investigation the following conclusions can be drawn:

- The GPSC algorithm can be applied to obtain the models that detect the faulty operation of the robot manipulator and show high-performance metrics;
- The investigation showed that with the application of OMs, the balance between class samples was reached, and using these types of datasets in GPSC generated high-performing models. So the conclusion is that dataset OMs have some influence on the classification accuracy of obtained results;
- Conducted research demonstrating that by using a balanced dataset with the SMOTE method in the GPSC algorithm with RSHVs and 5FCV, the best SEs in terms of high mean evaluation metric values with low standard deviation can be obtained. When the aforementioned SEs were applied to the initial imbalanced dataset the results of EMs slightly deviate from those obtained on the SMOTE dataset.
- The GPSC algorithm as applied procured a set of the best SEs that can be used to obtain a robust solution;

- This investigation also showed that not all variables are necessary to detect the faulty operation of a robotic manipulator. In this case, a total of 26 input variables are not required which is a great reduction in the experimental measurement of these variables.

  The main pros of the proposed method are:

- The entire GPSC model does not have to be stored. No matter how long the equation is it still requires lower computational resources than the entire CNN or DNN. The aforementioned CNN or DNN can not be simply transformed into a form of SE.
- The generated SEs with GPSC in some cases do not require all dataset input variables. However, other machine learning methods require all input variables that were used to train them.

  The main cons of the proposed method are:

- The development of the RSHVs method is a time-consuming process that requires changing each hyperparameter value and running GPSC execution to investigate its influence on the performance of the algorithm;
- The ParsCoef is the most sensitive for tuning. A small change in its value can have a great impact on the GPSC performance.

  Future work will be focused on creating the original dataset, preferably balanced. The idea is to apply the same procedure in an experimental study to validate the mathematical equations obtained using the same procedure described in this paper. Besides that, the dataset which was used in this investigation will be subjected to a synthetic data generation process to try artificially create samples of the original dataset and after that perform oversampling. Synthetic data generation, in this case, is mandatory, especially for classes with an extremely small number of samples. Hopefully, this procedure will contribute to high classification accuracy in the multi-class problem.

**Author Contributions:** Conceptualization, N.A. and I.L.; data curation, N.A. and S.B.Š.; formal analysis, I.L. and M.G.; funding acquisition, N.A. and I.L.; investigation, S.B.Š., and M.G.; methodology, N.A. and S.B.Š.; project administration, N.A. and I.L.; resources, I.L.; resources, I.L.; software, N.A. and M.G.; supervision, N.A. and I.L.; validation, I.L.; visualization, S.B.Š.; writing—original draft, N.A. and S.B.Š.; writing—review and editing, I.L. and M.G. All authors have read and agreed to the published version of the manuscript.

**Funding:** This research received no external funding.

**Institutional Review Board Statement:** The institutional review board statement is not applicable.

**Informed Consent Statement:** The informed consent statement is not applicable.

**Data Availability Statement:** Publicly available repository located at https://archive.ics.uci.edu/ml/datasets/Robot+Execution+Failures, accessed on 1 December 2022.

**Acknowledgments:** This research has been (partly) supported by University of Rijeka scientific grants uniri-mladi-technic-22-61, uniri-mladi-technic-22-57, uniri-tehnic-18-275-1447, the CEEPUS network CIII-HR-0108, project CEKOM under the grant KK.01.2.2.03.0004, European Regional Development Fund under the grant KK.01.1.1.01.0009 (DATACROSS), and Erasmus+ project WICT under the grant 2021-1-HR01-KA220-HED-000031177.

**Conflicts of Interest:** The authors of this paper declare no conflict of interest.

## Appendix A. Dataset Additional Information

*Appendix A.1. Dataset Statistics and GPSC Variable Representation*

In Table A1 the result of dataset statistical analysis is shown as well as the GPSC variable representation.

**Table A1.** Dataset statistics and GPSC variable representation.

| Dataset Variable | Mean | Std | Min | Max | GPSC Variable | Dataset Variable | Mean | Std | Min | Max | GPSC Variable |
|---|---|---|---|---|---|---|---|---|---|---|---|
| Class | 0.721382 | 0.448804 | 0 | 1 | $y$ | $T_8^x$ | −10.9136 | 70.98129 | −524 | 400 | $X_{45}$ |
| $F_1^x$ | 5.429806 | 54.7229 | −254 | 353 | $X_0$ | $T_8^y$ | −5.7905 | 66.34445 | −492 | 433 | $X_{46}$ |
| $F_1^y$ | 1.045356 | 44.96064 | −338 | 219 | $X_1$ | $T_8^z$ | −4.35421 | 17.17883 | −150 | 64 | $X_{47}$ |
| $F_1^z$ | −37.7624 | 369.2985 | −3617 | 361 | $X_2$ | $F_9^x$ | −1.57451 | 47.21401 | −389 | 339 | $X_{48}$ |
| $T_1^x$ | −5.23974 | 117.2394 | −450 | 686 | $X_3$ | $F_9^y$ | −2.28942 | 36.62957 | −343 | 190 | $X_{49}$ |
| $T_1^y$ | 7.358531 | 111.824 | −286 | 756 | $X_4$ | $F_9^z$ | −62.5702 | 403.8498 | −2792 | 151 | $X_{50}$ |
| $T_1^z$ | −1.77322 | 25.55332 | −137 | 149 | $X_5$ | $T_9^x$ | −8.7581 | 70.60608 | −567 | 410 | $X_{51}$ |
| $F_2^x$ | 3.481641 | 48.37184 | −246 | 337 | $X_6$ | $T_9^y$ | −7.83369 | 62.94308 | −487 | 437 | $X_{52}$ |
| $F_2^y$ | −0.1987 | 39.63278 | −360 | 205 | $X_7$ | $T_9^z$ | −2.77322 | 13.33397 | −83 | 88 | $X_{53}$ |
| $F_2^z$ | −58.1361 | 399.9206 | −3261 | 146 | $X_8$ | $F_{10}^x$ | 0.416847 | 34.27027 | −248 | 338 | $X_{54}$ |
| $T_2^x$ | −7.9892 | 86.15365 | −467 | 605 | $X_9$ | $F_{10}^y$ | −3.30022 | 36.81007 | −353 | 188 | $X_{55}$ |
| $T_2^y$ | −7.49028 | 64.08104 | −271 | 261 | $X_{10}$ | $F_{10}^z$ | −51.7041 | 340.3173 | −2788 | 89 | $X_{56}$ |
| $T_2^z$ | −1.3067 | 15.80375 | −69 | 135 | $X_{11}$ | $T_{10}^x$ | −9.11015 | 68.56809 | −563 | 408 | $X_{57}$ |
| $F_3^x$ | 2.598272 | 42.13289 | −247 | 331 | $X_{12}$ | $T_{10}^y$ | −5.28726 | 62.68289 | −502 | 462 | $X_{58}$ |
| $F_3^y$ | −2.32829 | 42.9849 | −367 | 276 | $X_{13}$ | $T_{10}^z$ | −3.19654 | 12.09228 | −89 | 97 | $X_{59}$ |
| $F_3^z$ | −61.5961 | 379.1127 | −3281 | 132 | $X_{14}$ | $F_{11}^x$ | −1.35853 | 65.97879 | −492 | 448 | $X_{60}$ |
| $T_3^x$ | −7.47084 | 85.93531 | −535 | 620 | $X_{15}$ | $F_{11}^y$ | −7.42765 | 51.87442 | −364 | 185 | $X_{61}$ |
| $T_3^y$ | −2.60691 | 74.5475 | −427 | 476 | $X_{16}$ | $F_{11}^z$ | −110.994 | 510.0633 | −3234 | 94 | $X_{62}$ |
| $T_3^z$ | −1.33693 | 15.03959 | −66 | 144 | $X_{17}$ | $T_{11}^x$ | −8.0108 | 78.76645 | −558 | 404 | $X_{63}$ |
| $F_4^x$ | 3.153348 | 39.23666 | −249 | 337 | $X_{18}$ | $T_{11}^y$ | −7.22678 | 75.12606 | −576 | 454 | $X_{64}$ |
| $F_4^y$ | −1.11447 | 45.40036 | −364 | 354 | $X_{19}$ | $T_{11}^z$ | −4.47948 | 22.87002 | −199 | 81 | $X_{65}$ |
| $F_4^z$ | −67.635 | 428.3144 | −3292 | 107 | $X_{20}$ | $F_{12}^x$ | −4.43844 | 88.33604 | −883 | 460 | $X_{66}$ |
| $T_4^x$ | −6.33693 | 84.95088 | −495 | 567 | $X_{21}$ | $F_{12}^y$ | −8.48812 | 55.64501 | −364 | 187 | $X_{67}$ |
| $T_4^y$ | −3.24406 | 75.36126 | −633 | 464 | $X_{22}$ | $F_{12}^z$ | −134.477 | 575.1008 | −3451 | 179 | $X_{68}$ |
| $T_4^z$ | −1.76026 | 17.14544 | −88 | 161 | $X_{23}$ | $T_{12}^x$ | −3.31102 | 105.7364 | −540 | 1016 | $X_{69}$ |
| $F_5^x$ | 0.717063 | 43.43762 | −251 | 351 | $X_{24}$ | $T_{12}^y$ | −5.80994 | 89.37178 | −568 | 458 | $X_{70}$ |
| $F_5^y$ | 0.667387 | 52.38475 | −368 | 438 | $X_{25}$ | $T_{12}^z$ | −4.75378 | 26.98989 | −233 | 86 | $X_{71}$ |
| $F_5^z$ | −79.6004 | 479.6485 | −3348 | 107 | $X_{26}$ | $F_{13}^x$ | −5.47516 | 89.50921 | −851 | 462 | $X_{72}$ |
| $T_5^x$ | −12.0842 | 105.922 | −824 | 536 | $X_{27}$ | $F_{13}^y$ | −7.26134 | 53.43668 | −352 | 181 | $X_{73}$ |
| $T_5^y$ | −8.77754 | 85.63418 | −725 | 406 | $X_{28}$ | $F_{13}^z$ | −140.566 | 586.8129 | −3275 | 126 | $X_{74}$ |
| $T_5^z$ | −2.21166 | 19.82233 | −128 | 201 | $X_{29}$ | $T_{13}^x$ | −7.3067 | 83.26654 | −516 | 400 | $X_{75}$ |
| $F_6^x$ | −0.55508 | 38.56243 | −262 | 324 | $X_{30}$ | $T_{13}^y$ | −10.2613 | 82.56735 | −567 | 471 | $X_{76}$ |
| $F_6^y$ | −2.2635 | 40.62626 | −320 | 254 | $X_{31}$ | $T_{13}^z$ | −4.15767 | 19.97507 | −197 | 93 | $X_{77}$ |
| $F_6^z$ | −57.8942 | 417.1217 | −3051 | 418 | $X_{32}$ | $F_{14}^x$ | −1.66307 | 69.58861 | −480 | 460 | $X_{78}$ |
| $T_6^x$ | −3.56371 | 89.31248 | −672 | 747 | $X_{33}$ | $F_{14}^y$ | −5.21598 | 48.83071 | −343 | 212 | $X_{79}$ |
| $T_6^y$ | −2.42549 | 67.07047 | −468 | 389 | $X_{34}$ | $F_{14}^z$ | −133.618 | 559.7376 | −3226 | 92 | $X_{80}$ |
| $T_6^z$ | −1.60259 | 23.169 | −162 | 244 | $X_{35}$ | $T_{14}^x$ | −7.85745 | 87.05328 | −527 | 531 | $X_{81}$ |
| $F_7^x$ | −2.69762 | 41.83346 | −389 | 338 | $X_{36}$ | $T_{14}^y$ | −7.89201 | 78.53241 | −600 | 466 | $X_{82}$ |

<div align="center">

**Table A1.** *Cont.*

</div>

| Dataset Variable | Mean | Std | Min | Max | GPSC Variable | Dataset Variable | Mean | Std | Min | Max | GPSC Variable |
|---|---|---|---|---|---|---|---|---|---|---|---|
| $F_7^y$ | −4.25918 | 38.11985 | −382 | 192 | $X_{37}$ | $T_{14}^z$ | −3.67819 | 20.93429 | −248 | 101 | $X_{83}$ |
| $F_7^z$ | −62.5767 | 440.9198 | −3557 | 126 | $X_{38}$ | $F_{15}^x$ | −3.2419 | 60.06207 | −497 | 342 | $X_{84}$ |
| $T_7^x$ | −4.06695 | 71.7687 | −547 | 472 | $X_{39}$ | $F_{15}^y$ | −3.55076 | 49.50523 | −343 | 242 | $X_{85}$ |
| $T_7^y$ | −6.91577 | 63.1772 | −429 | 601 | $X_{40}$ | $F_{15}^z$ | −104.626 | 483.9425 | −2955 | 95 | $X_{86}$ |
| $T_7^z$ | −2.99352 | 13.10753 | −91 | 83 | $X_{41}$ | $T_{15}^x$ | −9.93521 | 87.94631 | −599 | 462 | $X_{87}$ |
| $F_8^x$ | −1.47732 | 41.89043 | −262 | 340 | $X_{42}$ | $T_{15}^y$ | −9.946 | 76.72955 | −646 | 466 | $X_{88}$ |
| $F_8^y$ | −1.99352 | 37.24492 | −331 | 190 | $X_{43}$ | $T_{15}^z$ | −2.90281 | 14.6582 | −91 | 108 | $X_{89}$ |
| $F_8^z$ | −57.879 | 389.6659 | −2795 | 97 | $X_{44}$ | | | | | | |

## Appendix B. The Best SEs

The set of best SEs that were obtained on the dataset balanced with SMOTE OM.

$$
\begin{aligned}
y_1 =\ & 2.\sin\!\Big(\big(1.4\log(\min(\sin(X_{81}),X_{44})) + \tan(\tan(|X_{81}|)) + 1.4\log(X_1) + 2\sqrt[3]{X_{16}} + \sin(\sqrt[3]{X_{40}}) \\
& + 1.4\log(X_{48}) + 2\sin(\sqrt[3]{X_{60}}) + 1.4\log(X_{78})\big)^{\frac{1}{3}}\Big) + 2.\sin\!\Big(\big(1.4\log(\min(\sin(X_{81}),X_{44})) + 1.4\log(X_1) \\
& + \sin(X_{11}+X_{12}) + 1.4\log(\log(X_{12})) + 2\sqrt[3]{X_{16}} + \sin(\sqrt[3]{X_{40}}) + 1.4\log(X_{48}) + \sin(\sqrt[3]{X_{60}}) + 1.4\log(X_{78})\big)^{\frac{1}{3}}\Big) \\
& + 2.\sqrt[3]{|\log(X_{12})| + 1.4\log(X_1) + \log(X_{12}) + \sqrt[3]{X_{16}} + \sin(\sqrt[3]{X_{40}}) + \sin(\sqrt[3]{X_{48}}) + \sin(\sqrt[3]{X_{60}}) + 1.4\log(\log(X_{78}))} \\
& + 2.\sqrt[3]{1.4\log(X_1) + 2\log(X_{12}) + \sqrt[3]{X_{16}} + \sin(\sqrt[3]{X_{40}}) + \sin(\sqrt[3]{X_{48}}) + \sin(\sqrt[3]{X_{60}}) + 1.4\log(\log(X_{78}))} \\
& + 2.\sin(\sqrt[3]{X_1 + X_{12} + 2\sin(\sqrt[3]{X_{40}}) + X_{77} + 1.4\log(X_{78})}) \\
& + 2.\sin(\sqrt[3]{X_1 + \sqrt[3]{X_{40}} + \sqrt[3]{\sin(\sqrt[3]{X_{40}})} + X_{77} + \tan(1.4\log(X_{83}))}) + 2.88539\log(X_1) + \sin(X_{11}+X_{12}) \\
& + 2.\sin(\sqrt[3]{X_{11}+X_{12}}) + 11.5416\log(\log(X_{12})) \\
& + 2.\sin(\sqrt[3]{2\sqrt[3]{X_{16}} + \frac{X_4}{X_{18}} + \sin(X_{40}) + 1.4\log(X_{48}) + 1.4\log(1.4\log(X_{48})) + \sqrt[3]{X_{60}}}) \\
& + 2.\sin(\sqrt[3]{2\sqrt[3]{X_{16}} + \frac{X_4}{X_{18}} + \sin(\sqrt[3]{X_{40}}) + 1.4\log(X_{48}) + 1.4\log(1.4\log(X_{48})) + \sqrt[3]{X_{60}}}) + 2.\sqrt[3]{X_{16}} + 2.\sqrt[3]{\frac{X_{56}}{X_{18}}} \\
& + 2.88539\log(X_{48}) + 2.\sin(\sqrt[3]{X_{60}}) + 2.88539\log(X_{78})
\end{aligned} \tag{A1}
$$

$$
\begin{aligned}
y_2 =\ & \min(1.4\log(X_0 + \tan(\log(X_{88}))), \min(1.4\log(X_0 + \tan(\log(X_{88}))), 1.4\log(\max(X_{28}, 1.2\sqrt{\log(\sqrt{X_{28}})}) \\
& - \min(X_{80}, \sin(X_{47}))) + 1.4\log(1.4\log(\max(X_{30}, X_4))) + \min(1.4\log(0.4 X_{25}\log(\sqrt{X_0})) \\
& + 0.4\log(\tfrac{X_{15}}{X_{14}}) + \log(\tfrac{X_{77}}{X_{39}}), 1.4\log(\tan(X_{63}))) + 1.4\log(\cos(\tan(\log(X_{88})) - 1.\min(X_{80}, \sin(X_{47})))) \\
& + 1.4\log(1.4\log(1.4\log(1.4\log(X_0))) + \sin(X_0) - 1.\sin(X_{47})) + 1.4\log(1.4\log(\sqrt{X_0})) \\
& + 0.4\log(\tfrac{X_{15}}{X_{14}}) + \log(X_{19}) + 0.4\log(X_{30}) + 0.4\log(\cos(X_{39})) + 1.4\log(X_{49}) + 3.\sqrt[3]{X_{52}} \\
& + 1.4\log(1.4\log(\sqrt[3]{X_{52}})) + \log(X_{73}) + \min(1.4\log(\sqrt{X_0} + \sin(X_{16})), 1.4\log(\max(1.4\log(\sqrt{X_0}), \\
& \log(X_{19})) - \sin(X_{47})) + 1.4\log(1.4\log(1.4\log(\sqrt{X_0})) + 0.4\log(0.4\log(X_{16}))) + \log(X_{18}) \\
& + 0.4\log(\cos(X_{45})) + \sqrt[3]{X_{52}} + 0.4\log(X_{78})) + \sqrt{\max(X_0, X_{30}, X_{44} X_{58}) - \min(X_{80}, \sin(X_4))} \\
& + \sqrt{\max(X_{18} X_{44}, 1.4\log(X_{49})) - 1.\min(X_{80}, \sin(X_{47}))} + \min(1.4\log(0.4 X_{25}\log(\sqrt{X_0})) \\
& + 0.4\log(\tfrac{X_{15}}{X_{14}}) + \log(\tfrac{X_{77}}{X_{39}}), 1.4\log(\tan(X_{63}))) + \sqrt{\log(X_{19}) - 1.\min(X_{80} - 1.X_7, 1.4\log(\sqrt{X_0}))}
\end{aligned} \tag{A2}
$$

$$
\begin{aligned}
&+ \quad \min(\log(X_{19}), 1.4\log(1.4\log(\sqrt{X_{28}}) - 1.\min(X_{80}, \sin(X_{47}))) + 1.4\log(X_{24}) + \sqrt[3]{X_{52}}) \\
&+ \quad 1.4\log(\cos(\tan(\log(X_{88}))) - 1.\min(X_{80}, \sin(X_{47}))) + 1.4\log(1.4\log(1.4\log(1.4\log(X_0)))) \\
&+ \quad \sin(X_0) - 1.\sin(X_{47})) + 1.4\log(1.4\log(\sqrt{X_0})) + 3.\log(X_{19}) + 0.4\log(0.4\log(\sin(X_3))) \\
&+ \quad 0.4\log(X_{30}) + 0.4\log(\cos(X_{39})) + 0.4\log(\cos(X_{45})) + 1.4\log(X_{49}) + 2.\sqrt[3]{X_{52}} \\
&+ \quad 1.4\log(1.4\log(\sqrt[3]{X_{52}})) + 0.4\log(\cos(\sqrt[3]{X_{52}})) + \log(X_{73})) + \min(1.4\log(\sqrt{X_0} + \sin(X_{16})), \\
&\qquad 1.4\log(\max(1.4\log(\sqrt{X_0}), \log(X_{19})) - 1.\sin(X_{47})) + 1.4\log(1.4\log(1.4\log(\sqrt{X_0})))) \\
&+ \quad 0.4\log(0.4\log(X_{16})) + \log(X_{18}) + 0.4\log(\cos(X_{45})) + \sqrt[3]{X_{52}} + 0.4\log(X_{78})) \\
&+ \quad \sqrt{\max(X_0, X_{30}, X_{44}X_{58}) - 1.\min(X_{80}, \sin(X_4))} + \sqrt{\max(X_{18}X_{44}, 1.4\log(X_{49})) - 1.\min(X_{80}, \sin(X_{47}))} \\
&+ \quad \sqrt{\log(X_{19}) - 1.\min(X_{80} - 1.X_7, 1.4\log(\sqrt{X_0}))} + \min(\log(X_{19}), 1.4\log(1.4\log(\sqrt{X_{28}}) \\
&- \quad 1.\min(X_{80}, \sin(X_{47}))) + 1.4\log(X_{24}) + \sqrt[3]{X_{52}}) + 2.\log(X_{19}) \\
&+ \quad 0.4\log(0.4\log(\sin(X_3))) + 0.4\log(\cos(X_{45})) + 0.4\log(\cos(\sqrt[3]{X_{52}}))
\end{aligned}
$$

$$
\begin{aligned}
y_3 = \ & 6(\sec(\cos(X_0))(\Big(2.4\cot(\cos(\cos(\tan(X_{82}))))(2.4(\sec(\cos(X_0))(\Big(2.4\cot(\cos(\cos(\tan(X_{82}))))(\sec(\cos(X_0)) \\
& (\cot(\cos(\cos(\log(X_0))))(\cot(\cos(\cos(\log(X_0))))(1.4(\max(\log(X_0), 2.4(\sec(\cos(X_0))(\cot(\cos(\cos(\log(X_0)))) \\
& (\cot(\cos(\cos(\log(\cos(\log(X_0)))))))(1.4(\max(X_{70}, X_{84}, \log(X_0))) + \cos(|\sqrt[3]{X_{86}}|)) + \log(\max(X_{28}, X_{36}, \\
& \log(\max(X_{18}, X_{36}, X_{70}, \log(X_0), \frac{\cos(\max(\cos(\log(X_{43})), \log(X_{43}))) + 1.4(\cos(\log(X_0)))}{X_{28}}, \\
& \log(\max(\sqrt{X_{59}}, X_{40}X_{68}, X_{70})))))))) + \log(\log(\max(X_{70}, \log(X_0), 3.55\log(X_{49})))) + \log(\max(X_{28}, X_{36}, \\
& \log(X_{36}))))))) + \cos(|\sqrt[3]{X_{86}}|)) + \log(\max(X_{28}, X_{36}, \log(\max(-4485.04, X_{18}, X_{36}, X_{70}, \log(X_0), \log(\max(X_{40}X_{68}, \\
& X_{69}, X_{70})))))))) + \log(\log(\max(X_{70}, \log(X_0), 3.55\log(X_{49})))) + \log(\max(X_{28}, \log(\max(X_{70}, \log(X_0), 2.4(\log(\max(X_{28}, \\
& X_{36}, \log(\max(X_{40}X_{68}, X_{70}, X_{81}))) + \frac{\cos(|\sqrt[3]{X_{86}}|)}{|\cos(\cos(\log(\cos(\log(X_0)))))|}))))))) + \log(\max(X_{28}, X_{36}, \log(\max(X_{70}, \\
& \cos(\log(X_{43})), \log(X_0), \log(X_{43}))))\Big)\Big/\Big(|\cos(\cos(\log(\cos(\log(X_0)))))| + \log(\log(\cos(\log(X_0)))) + \log(\max(X_{28}, \\
& X_{36}, \log(\max(X_{28}, X_{36})))))) + \log(\max(X_{28}, \log(\max(X_{70}, \log(X_0), 2.4(\log(\max(X_{28}, X_{36}, \log(\max(X_{40}X_{68}, \\
& X_{70}, X_{81}))) + \frac{\cos(|\sqrt[3]{X_{86}}|)}{|\cos(\cos(\log(\cos(\log(X_0)))))|}))))))) + \log(\max(X_{28}, X_{36}, \log(\max(X_{70}, \cos(\log(X_{43})), \log(X_0), \\
& \log(X_{43})))))\Big)\Big/\Big(|\cos(\cos(\log(\cos(\log(X_0)))))| + \log(\log(\cos(\log(X_0)))) + \log(\max(X_{28}, X_{36}, \log(\log(X_0)))))
\end{aligned} \tag{A3}
$$

$$
\begin{aligned}
y_4 = \ & \max(X_{22}, \max(X_{22}, X_{88}, \max(X_{22}, X_{76}, X_{88}, 1.4\log(\min(X_{80}, 0.4\log(\max(X_0, 0.4\log(X_{20})))))), \max(X_{16}, X_{18}, \\
& X_{22}, X_{25}, X_{28}, X_{59}, X_{88}, \max(X_{18}, X_{59}, X_{88}, 1.4\log(\min(X_{80}, 0.4\log(\log(0.4\log(\max(\frac{X_{51}}{X_{21}}, 1.4\log(|\log(\max(X_0, \\
& 0.4\log(X_{20})))|)))) + |\tan(X_{59})|))))) + \max(\sqrt{X_1}, X_{88}, X_{27} + 1.4\log(0.4\log(\log(X_8))), \sqrt{X_{49}}|X_0|\min(X_{77}, X_{79}))) \\
&+ \quad \max(X_{22}, X_{76}, X_{88}, 1.4\log(\min(X_{80}, 0.4\log(\max(X_0, 0.4\log(X_{20})))))), \max(X_{28}, 1.4\log(\min(X_2, X_{50} \\
&+ \quad X_9, 0.4\log(0.4\log(X_{20})))), \max(X_{22}, X_{76}, 1.4\log(\min(X_2, X_{80}, 0.4\log(\log(0.4\log(\max(\frac{X_{51}}{X_{21}}, 0.4\log(\sqrt[3]{X_{27}})) \\
&+ \quad |\log(0.4\log(X_{20}))|))))) + \max(X_{16}, X_{18}, X_{22}, X_{25}, X_{28}, X_{40}, X_{46}, X_{59}, \sin(X_{44}))) + 1.4\log(\min(X_2, \\
&\qquad 0.4\log(\log(0.4\log(\max(\frac{X_{51}}{X_{21}}, X_{88})) + |\log(0.4\log(X_{20}))|)))) + 1.4\log(\min(X_{80}, 0.4\log(\log(\tan(X_{59}) \\
&+ \quad 0.4\log(\frac{X_{51}}{X_{21}}))))))) + \max(X_{16}, X_{18}, X_{22}, X_{25}, X_{28}, X_{40}, X_{46}, X_{59}, \sin(X_{44}))) + 1.4\log(\min(X_2, X_{80}, \\
&\qquad 0.4\log(\log(0.4\log(\max(X_0, 0.4\log(X_{20})))))))), \sin(X_{44})) + \max(X_{28}, 1.4\log(\min(X_2, X_{50} + X_9, \\
&\qquad 0.4\log(0.4\log(X_{20})))), \max(X_{22}, X_{76}, 1.4\log(\min(X_2, X_{80}, 0.4\log(\log(0.4\log(\max(\frac{X_{51}}{X_{21}}, 0.4\log(\sqrt[3]{X_{27}})) \\
&+ \quad |\log(0.4\log(X_{20}))|))))) + \max(X_{16}, X_{18}, X_{22}, X_{25}, X_{28}, X_{40}, X_{46}, X_{59}, \sin(X_{44}))) + 1.4\log(\min(X_2, \\
&\qquad 0.4\log(\log(0.4\log(\max(\frac{X_{51}}{X_{21}}, X_{88})) + |\log(0.4\log(X_{20}))|)))) + 1.4\log(\min(X_{80}, 0.4\log(\log(\tan(X_{59}) \\
&+ \quad 0.4\log(\frac{X_{51}}{X_{21}}))))))) + \max(X_{18}, X_{59}, X_{88}, 1.4\log(\min(X_{80}, 0.4\log(\log(0.4\log(\max(\frac{X_{51}}{X_{21}}, 1.4\log(|\log(\max(X_0, \\
&\qquad 0.4\log(X_{20})))|)))) + |\tan(X_{59})|))))) + \max(\sqrt{X_1}, X_{88}, X_{27} + 1.4\log(0.4\log(\log(X_8))), \sqrt{X_{49}}|X_0|\min(X_{77}, X_{79})))
\end{aligned} \tag{A4}
$$

$$+ \quad 1.4\log(\min(X_2, X_{80}, 0.4\log(\log(0.4\log(\max(X_0, 0.4\log(X_{20}))))))) + 1.4\log(0.4\log(\max(X_0, 0.4\log(X_{20}))))$$

$$+ \quad 1.4\log(0.4\log(0.4\log(X_{20})))$$

$$y_5 \quad = \quad \max(X_{16}, \max(X_{16}, \max(X_{16}, \max(X_{16}, \max(X_{46}, \left(1.4\log(\sin((\max(X_{16}, \tag{A5}$$

$$\frac{\log(X_{33})}{\max(1.4\log(1.20112\sqrt{\log(\sqrt[3]{X_{73}})}), \tan(0.4\log(X_{73})))})) \bigg/ \left(|X_{28}|\right))) + \max(X_{42}, \tan(1.4\log(1.4$$

$$\log(1.2\sqrt{\log(\sqrt{\min(X_{14}, X_{18})})})))))\right)^{\frac{1}{3}}) + 1.4\log(\max(X_{16}, \left(1.4\log(\sin(\sqrt{\max(X_0, X_{32})}))\right.$$

$$+ \quad \tan(1.4\log(0.65\sqrt{\log(X_{69})})))\big)\frac{1}{3}) + 1.4\log(\sqrt{\min(X_{14}, X_{18})})) + \max(X_{16}, \max(X_{16}, X_{46}, \max(X_{16}, X_{46},$$

$$\left(\max(X_{16}, X_{46}, \max(X_{16}, \left(\max(X_{16}, \tan(1.4\log(1.2\sqrt{\log(\sqrt{\min(X_3, X_{66})})}))) + \max(X_{16}, \right.$$

$$1.13\sqrt[3]{\log(\sin(\max(X_0, X_{32})))} + 1.4\log(\sin(\sqrt{X_{67}}))\big)^{\frac{1}{3}}) + 1.4\log(\sqrt{\min(X_{18}, X_{86})})\big)^{\frac{1}{3}}) + \max(X_{16},$$

$$\left(1.4\log(\max(X_{16}, \log(X_0), \sqrt{\max(X_{16}, \sqrt{\min(X_{18}, X_{74})}, \tan(1.4\log(1.4\log(\sqrt{X_{18}}))))})) + \max(X_{16}, \right.$$

$$\tan(1.4\log(1.4\log(\sqrt{X_{18}})))) + \max(X_{16}, \sqrt[9]{X_4})\big)^{\frac{1}{3}})) + \max(X_{24}, \tan(\tan(1.4\log(1.20112\sqrt{\log(X_{36})}))))))))$$

$$+ \quad \max(1.4\log(\sqrt{\min(X_{18}, X_{86})}), \tan(0.4\log(X_{73})))) + \max(X_{16}, \max(X_{16}, \left(\max(X_{16}, \max(X_{16}, \left(\max(X_{16}, \right.$$

$$1.13\sqrt[3]{\log(\sin(\sqrt{\max(X_0, X_{32})}))} + \sqrt[9]{X_4} + 1.4\log(\sin(1.4\log(X_{67})))\big)^{\frac{1}{3}}) + 1.4\log(\sqrt{\min(X_{18}, X_{86})})\big)^{\frac{1}{3}})$$

$$+ \quad \max(X_{16}, \left(\max(X_{16}, \max(X_{16}, X_{46}, \sqrt[3]{X_4}, \tan(1.4\log(\min(X_{14}, X_{18})))) + \max(X_{16}, 1.4\log(\sin(\frac{\log(X_{69})}{\log(X_{56})})))\right.$$

$$+ \quad 1.4\log(\frac{\log(X_{33})}{\log(X_{56})})) + \max(X_{16}, \tan(1.4\log(1.4\log(\sqrt{\min(X_{18}, X_{74})}))))\big)^{\frac{1}{3}})) + \max(X_{18}, \max(X_{16}, \tan(1.4$$

$$\log(\sqrt{\max(X_{16}, \tan(1.4\log(1.4\log(\sqrt{X_{18}}))))})))) + 1.4\log(\sin(\frac{\sqrt{\min(X_{18}, X_{74})}}{\log(X_{56})}))) + 1.4\log(1.4$$

$$\log(\sqrt{\min(X_{74}, \sqrt[3]{X_4})}))) + \max(1.4\log(\max(X_{24}, \tan(1.4\log(\max(X_{16}, \sqrt[3]{1.4\log(\sin(X_{14})) + \cos(\frac{X_{27}}{X_{18}})}))$$

$$+ \quad \max(X_{16}, \tan(1.4\log(X_{63})) + X_2)))), \tan(\tan(1.4\log(1.20112\sqrt{\log(\sqrt{X_{73}})}))))))$$

It should be noted that during GPSC executions mathematical functions division, natural logarithm, square root, and logarithms with base 2 and 10 are modified to avoid errors during execution. If the Equations (A1)–(A5) are used then the aforementioned mathematical functions must be applied which are defined in the following way:

- Division function

$$y_{DIV}(x_1, x_2) = \begin{cases} \frac{x_1}{x_2} & \text{if} |x_2| > 0.001 \\ 1 & \text{if} |x_2| < 0.001 \end{cases}. \tag{A6}$$

- Square root

$$y_{SQRT}(x) = \sqrt{|x|}, \tag{A7}$$

- Natural logarithm

$$y_{log}(z) = \begin{cases} \log(|z|) & \text{if} |z| > 0.001 \\ 0 & \text{if} |z| < 0.001 \end{cases} \tag{A8}$$

The Equation (A8) can be applied to $\log_2$, and $\log_{10}$ however log must be replaced with $\log_2$, and $\log_{10}$, respectively. The $z$, $x_1$, and $x_2$, in Equations (A6)–(A8) are arbitrary variable names.

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
