# Peer review of "Classification of Faults Operation of a Robotic Manipulator Using Symbolic Classifier"

_applsci, doi:10.3390/app13031962_

Round 1
Reviewer 1 Report
The paper presents an application of a symbolic classifier to a publicly available dataset to obtain symbolic expression as a mathematical equation. Where the objective is to detect the fault operation of a robotic manipulator with high accuracy.
As comments:
- The paper is well presented and the results are well discussed,
- The abstract contains a sufficient summary of the work;
- The paper deals with an interesting idea and relevant to the scope of the review and reflects sufficiently and clearly the topic of the journal;
- In the introduction, authors must add the organization of the paper (as a definition of the presented sections),
- In this paper, authors are based on a limited number of references (15 refs), so the contribution must be compared and referenced with the latest and novel papers.
- Do the authors have a comparison with other applications in the same subject? Please discuss.
Author Response
The authors want to thank the reviewer for his time and effort to give his constructive comments and suggestions which have greatly improved the manuscript's quality. The authors do hope that manuscript in this form is acceptable for publication. The answers to the comments are given below.
The paper presents an application of a symbolic classifier to a publicly available dataset to obtain symbolic expression as a mathematical equation. Where the objective is to detect the fault operation of a robotic manipulator with high accuracy.
As comments:
- The paper is well presented and the results are well discussed,
Answer: Thank you for the comment.
- The abstract contains a sufficient summary of the work;
Answer: Thank you for the comment.
- The paper deals with an interesting idea and relevant to the scope of the review and reflects sufficiently and clearly the topic of the journal;
Answer: Thank you for the comment.
- In the introduction, authors must add the organization of the paper (as a definition of the presented sections),
Answer: In the original and revised version of the manuscript the organization of the paper was located at the end of the Introduction section. Citing from the original and revised version of the manuscript:
“This paper consists of the following sections i.e. Materials and Methods, Results, Discussion, and Conclusions sections, respectively. In Materials and Methods, the research methodology is described as well as dataset description, dataset balancing methods, genetic programming-symbolic classifiers, random hyperparameter search with 5-fold cross-validation, evaluation metrics and computational resources. In the Results section, the best symbolic expressions obtained with the GPSC method obtained on each dataset variation are described, and the best symbolic expression among them was shown and evaluated on the original dataset. In the Discussion section, the dataset statistics, as well as obtained results are discussed. Finally, in the Conclusions section, the conclusions are drawn based on the hypotheses created in the Introduction section and the discussion created in the Discussion section.”
- In this paper, authors are based on a limited number of references (15 refs), so the contribution must be compared and referenced with the latest and novel papers.
Answer: The number of references was expanded from an initial 15 to 32. Special attention was focused on finding research papers that investigated the same dataset using different ML methods. From investigation, it was shown that non of the previous literature used Genetic Programming - Symbolic Classifier method. It should be noted that in the Introduction section, Table 1 was created where all papers that investigated the same dataset using different ML methods are listed. The table also contains the methods which were used to investigate this dataset and achieved classification accuracy. Table 1 was created to clearly present the previous literature in which this dataset was used.
- Do the authors have a comparison with other applications in the same subject? Please discuss.
Answer: In the revised version of the manuscript as suggested by all the reviewers the number of references was expanded from an initial 15 to 34 references. The introduction section was expanded and divided into two subsections. The first subsection is the previous literature overview and the last paragraphs of this subsection and Table 1 show the results that were obtained in other research papers using different ML methods on the same dataset that was used in this research. In the Results subsection entitled the same table is shown with one additional row in which the results of this research are presented. This Table was repeated to show the comparison of our results with previous research. After the Table and in the Discussion section the results were commented on.
Reviewer 2 Report
This paper presented a fault detection for robotic manipulator using symbolic classifier. The comments are given below.
1. The novelty of the paper has not been clearly stated. Methodologies such as the dataset balancing, genetic algorithm, are well-studied in the literature.
2. The proposed method is well-tested on an existing dataset. It is suggested to carry on the experimental study, preferably in real-time, on an actual robotic manipulator system to verify the effectiveness of the method.
3. The presentation and language shall be improved substantially, such as spelling, punctuation, indent, etc.
4. There are lots of redundant and repeated statement for example, in Abstract and in Section 2.1.
5. The abstract should be concise with key points only.
6. Section 2.2, the dataset [?], please correct it.
7. Table 1 and the working in Section 3.2 should be moved to the Appendix.
8. The subtitle for Table 2 is too long.
9. The description of of GA method should be shortened rather than the current lengthy one which is already well-known.
Author Response
The authors want to thank the reviewer for his time and effort to give his constructive comments and suggestions which have greatly improved the manuscript's quality. The authors do hope that manuscript in this form is acceptable for publication. The answers to the comments are given below.
This paper presented a fault detection for robotic manipulator using symbolic classifier. The comments are given below.
- The novelty of the paper has not been clearly stated. Methodologies such as the dataset balancing, genetic algorithm, are well-studied in the literature.
Answer: The novelty of this paper was the application of dataset balancing methods to balance the dataset and apply the genetic programming-symbolic classifier (not genetic algorithm (GA)) to obtain symbolic expression (mathematical equation) using which the robot fault operation can be detected with high classification accuracy. As seen in the original version of the manuscript the equations are obtained and can detect the robot fault operation with high classification accuracy.
To explain the novelty a little bit more with the application of the GPSC method symbolic expressions were obtained that can detect robot fault operation. The GPSC has some elements of GA but it is not the GA. The authors suggest to the reviewer further investigation of GPSC or Genetic programming -symbolic classifier and carefully read the Introduction section since the novelty was already stated in the original version of the manuscript.
However, the authors agree with the reviewer's suggestion, and the introduction section was modified i.e. two subsection titles were introduced and these are:
- Fault detection using Machine Learning Methods, and
- The definition of novelty, hypotheses, scientific contribution
Citing the novelty and the idea of this research from the subsection entitled “The Definition of novelty, hypotheses, the scientific contribution” from the revised version of the manuscript:
“The novelty of this paper is to implement a genetic programming-symbolic classifier (GPSC) algorithm to obtain symbolic expressions which can detect fault operation with high classification accuracy. The symbolic expressions are easier to implement in existing control systems of robotic manipulators since they require lower computational resources when compared to complex ML models such as CNN or DNN. So the idea of this paper is to show the procedure of how can GPSC be utilized to obtain symbolic expressions for the detection of robot manipulator fault operation with high classification accuracy.”
- The proposed method is well-tested on an existing dataset. It is suggested to carry on the experimental study, preferably in real-time, on an actual robotic manipulator system to verify the effectiveness of the method.
Answer: In this paper, the authors wanted to show that by using GPSC (not GA) with 5-fold cross-validation and random hyperparameter search method symbolic expressions can be obtained that can detect fault operation of the robot with high classification accuracy. Unfortunately, we do not have the required sensor equipment to obtain these measurements at the moment. However, we have included this idea in future work so it will definitely be one of the future steps.
- The presentation and language shall be improved substantially, such as spelling, punctuation, indent, etc.
Answer: Thank you for the comment. In the revised version of the manuscript, the language was improved, and spelling, punctuation, and indentation errors were corrected to the best of our knowledge.
- There are lots of redundant and repeated statement for example, in Abstract and in Section 2.1.
Answer: The abstract and section 2.1 have been rewritten. The authors do hope that in this form the abstract and Section 2.1 are ok. The authors have made an effort to reduce the number of redundant and repeated statements throughout the manuscript. By doing so the size of the manuscript has been slightly reduced.
Citing abstract from the revised version of the manuscript:
“In autonomous manufacturing lines, it is very important to detect the faulty operation of robot manipulators to prevent potential damage. In this paper, a genetic programming-symbolic classifier (GPSC) with random hyperparameter search and 5-fold cross-validation was applied to a publicly available dataset to obtain symbolic expression which can detect the fault operation of a robotic manipulator with high classification accuracy. The original dataset was reduced to a binary dataset (fault vs. normal operation) however, due to the class imbalance random oversampling, and SMOTE methods were applied. The quality of best symbolic expressions was based on the highest mean values of accuracy ($\overline{ACC}$), area under receiving operating characteristics curve ($\overline{AUC}$), $\overline{Precision}$, $\overline{Recall}$, and $\overline{F1-Score}$. The best symbolic expressions were obtained on SMOTE dataset with $\overline{ACC}$, $\overline{AUC}$, $\overline{Precision}$, $\overline{Recall}$, and $\overline{F1-Score}$ equal to 0.99, 0.99, 0.992, 0.9893, and 0.99, respectively. Finally, the best set of symbolic expressions was evaluated on the original dataset and the mean values of $\overline{ACC}$, $\overline{AUC}$, $\overline{Precision}$, $\overline{Recall}$, and $\overline{F1-Score}$ are equal to 0.9978, 0.998, 1.0, 0.997, and 0.998, respectively. The investigation showed that using described procedure symbolic expressions with high classification accuracy are obtained in the detection of fault operation of robotic manipulators.”
Citing Section 2.1 (Research Methodology) from the revised version of the manuscript:
“Since the original dataset is imbalanced the idea is to apply the dataset oversampling methods to equalize the dataset (random oversampling and SMOTE). The application of these methods produced two variations of the original dataset which were used in the GPSC algorithm with the random hyperparameter search and 5-fold cross-validation method to obtain symbolic expressions with high classification accuracy in the detection of fault operation of robot manipulator. The symbolic expressions obtained on each dataset variation were compared and the best symbolic expression is obtained based on classification performance and its size since the idea is to achieve the highest classification performance and the lower size of symbolic expression. Finally, the set of best symbolic expressions will be evaluated on the original dataset. The graphical presentation of the research methodology is shown in Figure 1.”
- The abstract should be concise with key points only.
Answer: The authors agree with the reviewer's comment and in the revised version of the manuscript the abstract was shortened to be concise as possible. Citing from the modified version of the manuscript. Citing abstract from the revised version of the abstract:
“In autonomous manufacturing lines, it is very important to detect the faulty operation of robot manipulators to prevent potential damage. In this paper, a genetic programming-symbolic classifier (GPSC) with random hyperparameter search and 5-fold cross-validation was applied to a publicly available dataset to obtain symbolic expression which can detect the fault operation of a robotic manipulator with high classification accuracy. The original dataset was reduced to a binary dataset (fault vs. normal operation) however, due to the class imbalance random oversampling, and SMOTE methods were applied. The quality of best symbolic expressions was based on the highest mean values of accuracy ($\overline{ACC}$), area under receiving operating characteristics curve ($\overline{AUC}$), $\overline{Precision}$, $\overline{Recall}$, and $\overline{F1-Score}$. The best symbolic expressions were obtained on SMOTE dataset with $\overline{ACC}$, $\overline{AUC}$, $\overline{Precision}$, $\overline{Recall}$, and $\overline{F1-Score}$ equal to 0.99, 0.99, 0.992, 0.9893, and 0.99, respectively. Finally, the best set of symbolic expressions was evaluated on the original dataset and the mean values of $\overline{ACC}$, $\overline{AUC}$, $\overline{Precision}$, $\overline{Recall}$, and $\overline{F1-Score}$ are equal to 0.9978, 0.998, 1.0, 0.997, and 0.998, respectively. The investigation showed that using described procedure symbolic expressions with high classification accuracy are obtained in the detection of fault operation of robotic manipulators.”
- Section 2.2, the dataset [?], please correct it.
Answer: The reference was missing in the original version of the manuscript and now in the revised manuscript version it does not. However, this paragraph was slightly modified. Citing the first paragraph from subsection 2.2:
“As stated in this research the publicly available dataset was used which can be downloaded from UCI Machine Learning Repository [18]. The dataset is well documented in [26,27] so in this subsection, only a brief dataset description is given. The dataset consists of a total of 463 data points and each of the data points D is shaped as:..”
- Table 1 and the working in Section 3.2 should be moved to the Appendix.
Answer: Table 1 and some parts of section 3.2 (formulas 9-13) are moved to the appendix section. The Table 1 is placed in Appendix A, and formulas are in Appendix B.
- The subtitle for Table 2 is too long.
Answer: The Subtitle for table 2 was shortened. The subtitle:The number of samples per class after application of dataset balancing methods.
- The description of of GA method should be shortened rather than the current lengthy one which is already well-known.
Answer: The GPSC is not entirely the same as GA. The GPSC uses elements of GA however it differently represents the solution than GA and for its execution it requires the dataset with input and output variables which is similar to Machine Learning.
So the authors think it is necessary to explain the GPSC since it has a mix of similarities between GA and ML.
However, as per your request the GPSR (not GA) description is shortened as much as possible. In the modified version of the GPSC description the following descriptions were omitted:
- description of GP treestructres,
- description of full and grow method i.e. basic elements of used ramped half-and-half method,
- crossover, subtree mutation, hoist mutation, and point mutation methods were removed, and
- parsimony coefficient is shortened.
Reviewer 3 Report
1. In the introduction, it is necessary to explain what the research problem is. Does it fill a specific gap in this field? Especially compared with the existing literature, what contribution does it make to the subject area? The introduction needs to be rewritten
2. Lack of literature review.。At present, only 15 articles are seriously insufficient. The author did not sort out and review the research in this field.
3. In addition, two articles about deep learning fault diagnosis are recommended to the author.
https://doi.org/10.3390/ijgi10100653
https://doi.org/10.1002/qre.2760
4. 2.2. Dataset description content is too redundant and should be placed in the appendix
5. Some formulas should be placed in the appendix, such as Formula 9 - Formula 13
6. Charts and tables need to be adjusted to make them more beautiful. Otherwise, it will affect readers' reading
7. in 112,there are errors“The dataset [? ] consists of a total of 463 data points”
8. The conclusion is not good enough.
Author Response
The authors want to thank the reviewer for his time and effort to give his constructive comments and suggestions which have greatly improved the manuscript's quality. The authors do hope that manuscript in this form is acceptable for publication. The answers to the comments are given below.
- In the introduction, it is necessary to explain what the research problem is. Does it fill a specific gap in this field? Especially compared with the existing literature, what contribution does it make to the subject area? The introduction needs to be rewritten
Answer: The authors agree with the reviewer’s comments and in the revised version of the manuscript the Introduction is rewritten. The used method fills the gap in this field since all methods used so far produced trained models which required substantial computational resources and this method generates symbolic expression (mathematical formula) which can be easily used and does not require a lot of computational resources to produce the results.
In the revised version of the manuscript, the Introduction consists of two subsections. In the first subsection entitled: “Fault detection using Machine Learning Methods” the literature overview was expanded as much as possible. First, the general implementation of ML methods for fault detection is described. Then the research papers are described which ML was applied to fault operation in robot manipulators. In the last paragraphs of this subsection, the notable research papers in which fault operation is detected using ML methods that used dataset as it was in this research are described. At the end of this subsection, the table with the list of these research papers is presented. Also, this table was used in the results section to compare our results with results from previous research.
In the second subsection entitled “The definition of novelty, hypotheses, the scientific contribution” the novelty, the idea in this paper, research hypotheses, scientific contributions, and the paper outline are clearly described.
- Lack of literature review. At present, only 15 articles are seriously insufficient. The author did not sort out and review the research in this field.
Answer: The Introduction is greatly expanded, and additional literature is included. The authors do hope that the introduction in this form is ok.
- In addition, two articles about deep learning fault diagnosis are recommended to the author.
https://doi.org/10.3390/ijgi10100653
https://doi.org/10.1002/qre.2760
Answer: The authors think that the connection between the suggested papers and this manuscript is vague. However, the authors have included these two papers in the manuscript (Introduction section). Citing from the revised version of the manuscript:
“In [10], the mass-customization with social internet of things (MC-SIOT) was proposed for effective connection between users and providers. The MC-SIOT can realize convenient information queries and clearly understand the user's intentions. The system can predict the changing relationships among different technical fields and help enterprise personnel to find technical knowledge. Combination of MC-SIOT with deep learning technology and digital twin technology to better maintain the operational state of the system.
In [11] chaotic back propagation (BP) neural network was utilized for the prediction of smart manufacturing information system reliability. The results showed that when SMIS fails, the failure behavior can easily lead SMIS into chaos through the propagation of an interdependent network.”
- 2.2. Dataset description content is too redundant and should be placed in the appendix
Answer: The authors believe that dataset description is a crucial element to give a detailed insight into the problem which was solved in this paper. However, the authors agree that some elements can be moved to the appendix so, the Table 2 for example is moved to appendix A.
- Some formulas should be placed in the appendix, such as Formula 9 - Formula 13
Answer: Since all 5 equations (symbolic expressions) are really long they are all moved to appendix B of the revised version of the manuscript.
- Charts and tables need to be adjusted to make them more beautiful. Otherwise, it will affect readers' reading
Answer: The charts and tables are improved in the revised version of the manuscript. Figure 5 showing GP tree was removed from revised version of the manuscript since other reviewers suggested that GPSC description is too long. Unfortunately, the Figure 4 is the same as in the original version since to our opinion it was the best way of showing the correlation between all input variables and the output “Class” variable.
- in 112,there are errors“The dataset [? ] consists of a total of 463 data points”
Answer: The reference was missing and now in the revised manuscript version it does not. However, the entire paragraph in which missing reference was located was rewritten. Citing from revised version of the manuscript:
“As stated in this research the publicly available dataset was used which can be downloaded from UCI Machine Learning Repository [18]. The dataset is well documented in [26,27] so in this subsection, only a brief dataset description is given. The dataset consists of a total of 463 data points and each of the data points D is shaped as:..”
- The conclusion is not good enough.
Answer: The conclusion is rewritten and improved. Besides the answers to hypotheses defined in the introduction section which is now expanded and future work, the advantages and disadvantages of the proposed method were described.
Round 2
Reviewer 2 Report
The authors have addressed my comments. However, the presentation can be further improved such as indent, linespace.
Equation 1 can be presented in a more concise way.
Figure 5 does not carry much useful information when presented in the current way.
Author Response
The authors want to thank the reviewer for his constructive comments and suggestions which greatly improved the manuscript's quality. The authors hope that the manuscript in this form is acceptable for publishing.
The authors have addressed my comments. However, the presentation can be further improved such as indent, and linespace.
Equation 1 can be presented in a more concise way.
Answer: Equation 1 is presented in a more concise way.
Figure 5 does not carry much useful information when presented in the current way.
Answer: Since Figure 5 does not carry useful information it was removed from the revised version of the manuscript. Figure 5 represented the training process of GPSC with random hyperparameter search with 5-fold cross-validation although it was mistakenly labeled as Table 4 in the last manuscript version. Since the process is previously described in describe the authors agree that Figure 5 was unnecessary.
Reviewer 3 Report
The author revised the article
It seems that putting the contents of the introduction into the literature review section will be more helpful to improve the article framework
Author Response
The authors want to thank the reviewer for his constructive comments and suggestions which greatly improved the manuscript's quality. The authors hope that the manuscript in this form is acceptable for publishing.
The author revised the article
It seems that putting the contents of the introduction into the literature review section will be more helpful to improve the article framework
Answer: The authors have removed the subsection title and merged the text before the subsection title with the text after the subsection title. In this way, the article framework was improved.